# Coastal flooding: impact of waves on storm surge during extremes. A case study for the German Bight

## Joanna Staneva<sup>1</sup>, Kathrin Wahle<sup>1</sup>, Wolfgang Koch<sup>1</sup>, Arno Behrens<sup>1</sup>, Luciana Fenoglio-Marc<sup>2</sup> and Emil V. Stanev<sup>1</sup>

1. Institute for Coastal Research, HZG, Max-Planck-Strasse 1, D-21502 Geesthacht, Germany

2. Institute of Geodesy and Geoinformation, University of Bonn, Nussallee 17, D- 53115 Bonn, Germany

Correspondence to: J. Staneva (joanna.Staneva@hzg.de)


#### Abstract

This study addresses the impact of wind, waves, tidal forcing and baroclinicity on the sea level of the German Bight during extremes storm events. The role of wave-induced processes, tides and baroclinicity is quantified, and the results are compared with in situ measurements and satellite data. A coupled high-resolution modelling system is used to simulate wind waves, the water level and the three-dimensional hydrodynamics. The models used are the wave model WAM and the circulation model GETM. The two-way coupling is performed via the OASIS3-MCT coupler. The effects of wind waves on sea level variability are studied, accounting for wavedependent stress, wave-breaking parameterization and wave-induced effects on vertical mixing. The analyses of the coupled model results reveal a closer match with observations than for the stand-alone circulation model, especially during the extreme storm Xaver in December 2013. The predicted surge of the coupled model is significantly enhanced during extreme storm events when considering wave-current interaction processes. The wave-dependent approach yields a contribution of more than 30% in some coastal areas during extreme storm events. The contribution of a fully three-dimensional model compared with a two-dimensional barotropic model showed up to 20% differences in the water level of the coastal areas of the German Bight during Xaver. The improved skill resulting from the new developments justifies further use of the coupled-wave and three-dimensional circulation models in coastal flooding predictions.


#### 1. Introduction

A challenging topic in coastal flooding research is the accurate prediction of sea surface elevation and wave heights. This is highly relevant over the European shelf, which is characterized by vast near-coastal shallow areas and a large near-coastal urban population. The increased demand to improve wave and storm predictions requires further development and improved representation of the physical processes in ocean models. The wind-induced surface stress over the ocean plays an important role in enhancing sea surface elevation (e.g., Flather, 2001). The importance of wind-wave–induced turbulence for the ocean surface layer was demonstrated by Davies *et al.* (2000), and it was further demonstrated for the bottom layer by Jones and Davies (1998) and for the wave-induced mixing by Babanin (2006) and Huang *et al.* (2011). Craig and Banner (1994) and Mellor (2003) suggested that surface waves can enhance mixing in the upper ocean. Qiao *et al.* (2004) developed a parameterization of wave-induced mixing with a circulation model. They found that wave-induced mixing can greatly enhance vertical mixing in the upper ocean.

Understanding the wave-current interaction processes is important for the coupling between the ocean, atmosphere and waves in numerical models. Longuet-Higgins and Stewart (1964) showed that wave-dissipation—induced gradients of radiation stress account for a transfer of wave momentum to the water column, changing the mean water level. The effects of waves on the lower marine atmospheric boundary layer have been demonstrated by a number of studies: Janssen (2004), Donelan *et al.* (2012), Fan *et al.* (2009), and for the light wind regimes: Veiga and Queiroz (2015); Sun *et al.*, (2015). The effects of wave-current interactions caused by radiation stress have also been addressed by Brown and Wolf (2009) and Wolf and Prandle (1999). A different approach, i.e., the vortex force formulation, was used by Bennis and Ardhuin (2011) and McWilliams *et al.* (2004), Kumar *et al.* (2012). The comparison of both methods by Moghimi *et al.* (2013) showed that the results are similar for longshore circulations, but radiation stress enhanced the offshore-directed transport in the wave shoaling regions. Many other studies addressed the role of the interaction between wind waves and circulation in the model simulations (Michaud *et al.*, 2012, Barbariol *et al.* 2013; Brown *et al.*, 2011; Katsafados *et al.*, 2016; Bolaños *et al.*, 2011, 2014, Röhrs et al, 2014).

Storm surges are meteorologically driven, typically by wind and atmospheric pressure. As shown by Holleman and Stacey (2014), an increasing water level decreases the frictional effects in the basin interior, which alters tidal amplification. Waves combined with higher water levels may break dykes, cause flooding, destroy construction and erode coasts (Pullen *et al.*, 2007). Waves can also modify the sediment dynamics (Grashorn *et al.*, 2015; Lettman *et al.*, 2009).

The German Bight is dominated by strong north-westerly winds and high waves due to Northeast Atlantic low-pressure systems (Rossiter, 1958; Fenoglio-Marc *et al.*, 2015). Extra-tropical cyclones in the area present a considerable hazard, especially in the shallow coastal Wadden Sea areas (Jensen and Mueller-Navarra, 2008). Coastal flooding can be caused by the combined effects of wind waves, high tides and storm surges in response to fluctuations in local and remote winds and atmospheric pressure. The role of these processes can be assessed using highresolution coupled models. However, in the frame of forecasting and climate modelling studies, the processes of wave and current interactions are not sufficiently exploited. In this study, we address the wave-current interaction to assess the impact of waves on the sea level of the German Bight during extremes. We quantify their individual and collective role and compare the model results with observational data that include various in situ and remote sensing measurements. The wave model (WAM), circulation model (GETM), study period and model experiments are presented in Section 2. The observational data are described in Section 3, followed by modeldata comparisons in Section 4. Finally, Section 5 addresses the effects of the different physical processes on the sea level variability, followed by concluding remarks in Section 6.

#### 2. Models

#### 2.1 Hydrodynamic Model



The circulation model used in this study is the General Estuarine Transport Model (GETM, Burchard and Bolding, 2002). The nested-grid model setup for the German Bight has a horizontal resolution of 1 km and 21  $\sigma$ -layers (Stanev *et al.*, 2011). GETM uses the k- $\varepsilon$ turbulence closure to solve for the turbulent kinetic energy k and its dissipation rate  $\varepsilon$ . The data for temperature, salinity, velocity and sea surface elevation at the open boundary are obtained from the coarser resolution (approximately 5 km and 21  $\sigma$ -layers) North Sea-Baltic Sea GETM model configuration (Staneva *et al.*, 2009). The sea surface elevation at the open boundary of the





outer (North Sea-Baltic Sea) model was prescribed using 13 tidal constituents obtained from the satellite altimetry via OSU Tidal Inversion Software (Egbert and Erofeeva, 2002). Both models were forced by atmospheric fluxes computed from bulk aerodynamic formulas. These formulas used model-simulated sea surface temperature, 2-m air temperature, relative humidity and 10-m winds from atmospheric analysis data. This information was derived from the COSMO-EU regional model operated by the German Weather Service (DWD; Deutscher Wetter Dienst), with a horizontal resolution of 7 km. River runoff data were provided by the German Federal Maritime and Hydrographic Agency (BSH; Bundesamt für Seeschifffahrt und Hydrographie).

#### 10 **2.2 Wave Model**

Ocean surface waves are described by the two-dimensional wave action density spectrum  $N(\sigma, \theta, \varphi, \lambda, t)$  as a function of the relative angular frequency  $\sigma$ , wave direction  $\theta$ , latitude  $\varphi$ , longitude  $\lambda$  and time t. The appropriate tool to solve the balance equation is the advanced third-generation spectral wave model WAM (WAMDI group, 1988, ECMWF, 2014). The use of the wave action density spectrum N is required if currents are taken into account. In that case, the action density is conserved, in contrast to the energy density, which is normally used in the absence of time-dependent water depths and currents. The action density spectrum is defined as the energy density spectrum  $E(\sigma, \theta, \varphi, \lambda, t)$  divided by  $\sigma$  observed in a frame moving with the ocean current velocity (Whitham, 1974, Komen *et al.*, 1994):

$$N(\sigma,\theta) = \frac{E(\sigma,\theta)}{\sigma}$$
 (1)

The wave action balance equation in Cartesian coordinates is given as:

$$\frac{\partial N}{\partial t} + \left(\mathbf{c}_{g} + \mathbf{U}\right) \nabla_{x,y} N + \frac{\partial c_{\sigma} N}{\partial \sigma} + \frac{\partial c_{\theta} N}{\partial \theta} = \frac{S_{wind} + S_{nl4} + S_{wc} + S_{bot} + S_{br}}{\sigma}$$
(2)


The first term on the left side of equation (2) represents the local rate of change of wave-energy density; the second term describes the propagation of wave energy in two-dimensional geographical space, where  $c_g$  is the group velocity vector and U is the corresponding current vector. The third term of the equation denotes the shifting of the relative frequency due to possible variations in depth and current (with propagation velocity  $c_{\sigma}$  in  $\sigma$  space). The last term on the left side of the equation represents depth-induced and current-induced refraction (with the propagation velocity  $c_{\theta}$  in  $\theta$  space). The term  $S = S(\sigma, \theta, \varphi, \lambda, t)$  on the right side of (2) is the net




source term expressed in terms of the action density. It is the sum of a number of source terms representing the effects of wave generation by wind  $(S_{wind})$  quadruplet nonlinear wave-wave interactions  $(S_{nl4})$ , dissipation due to white capping  $(S_{wc})$ , bottom friction  $(S_{bot})$  and wave breaking (S<sub>br</sub>). The current version of the third-generation wave model WAM Cycle 4.5.4 is an update of the former Cycle 4, which is described in detail in Komen et al. (1994) and Günther et al. (1992). The basic physics and numerics are maintained in the new release. The source function integration scheme is provided by Hersbach and Janssen (1999), and the updated source terms of Bidlot et al. (2007) and Janssen (2008) are incorporated. Depth-induced wave breaking (Battjes & Janssen, 1978) is included as an additional source function. The depth and/or current fields can be non-stationary. The wave models have the same resolution, and the model uses the same bathymetry and wind forcing as the GETM model. The boundary values of the North Sea model are taken from the operational regional wave model of the DWD, while the boundary values for the German Bight are taken from the North Sea model. The wave models run in shallow water mode, including depth refraction and wave breaking, and calculate the two-dimensional energy density spectrum at the active model grid points in the frequency/direction space. The solution of the WAM transport equation is provided for 24 directional bands at 15° each with the first direction being 7.5°, measured clockwise with respect to true north, and 30 frequencies logarithmically spaced from 0.042 Hz to 0.66 Hz at intervals of  $\Delta f/f = 0.1$ .

#### 20 **2.3 Coupled-wave circulation model implementation**

The implementation of the depth-dependent equations of the mean currents  $\boldsymbol{u}(\boldsymbol{x}, \boldsymbol{z}, \boldsymbol{t})$  in the presence of waves follows Mellor (2011). The momentum equation for an incompressible fluid is  $d\boldsymbol{u}/dt = \boldsymbol{F} - \nabla \delta \boldsymbol{p}$ , where  $\boldsymbol{F}$  is the sum of external forces (Coriolis, gravity, friction) and  $\nabla \delta \boldsymbol{p}$ 


is the pressure gradient, which includes the influence of wave motion on the mean current. Within the radiation stress formulation of Mellor (2011), the prognostic velocity  $\boldsymbol{u}$  is related to the Eulerian wave-averaged velocity. Using linear wave theory and accounting for the second-order terms of the wave height, the equation of motion is:

$$\frac{\partial \langle \mathbf{u} \rangle}{\partial t} = \langle \mathbf{F} \rangle - \langle \mathbf{u} \rangle \cdot \frac{\partial \langle \mathbf{u} \rangle}{\partial \mathbf{x}} - \frac{\partial}{\partial \mathbf{x}} \cdot \mathbf{S}, \tag{3}$$

where the angle brackets denote averaging over the wave period, and S is the radiation stress tensor:

$$\mathbf{S} = E\left(\frac{c_f}{c_g}\left[\frac{\mathbf{k}\otimes\mathbf{k}}{k^2} + \delta\right]\frac{\sinh 2kh + 2kh}{\sinh 2kD + 2kD} - \delta\frac{\cosh 2kh - 1}{4\sinh^2 kD}\right),\tag{4}$$

where  $E = 1/16gH_s$  is the wave energy, **k** is the wave vector, and  $h = D(1 + \xi)$  is the local depth of layer  $\xi$ . Thus, the divergence of the radiation stress is the only force related to waves in the momentum equations. The equation for kinetic energy, which is derived from the momentum equation by multiplication with the velocity vector, is:





$$\frac{\partial E_{kin}}{\partial t} = \langle \mathbf{F} \rangle \cdot \langle \mathbf{u} \rangle - \langle \mathbf{u} \rangle \cdot \frac{\partial E_{kin}}{\partial \mathbf{x}} - \frac{\partial}{\partial \mathbf{x}} \cdot \mathbf{S} \cdot \langle \mathbf{u} \rangle, \tag{5}$$

where the gradients in wave energy (*i.e.*, dissipation due to wave breaking) may lead to increased surface elevation (wave setup).

The wave state information required to account for the divergence of the radiation stress in the GETM momentum equations is provided by WAM. The dissipation source functions (wave breaking and white capping, as well as bottom dissipation) estimated by the wave model WAM are also used in the turbulence module of GOTM. These data are used to specify the boundary conditions for the dissipation of the turbulent kinetic energy and the vorticity due to wave breaking and bottom friction (Pleskachevsky *et al.*, 2011) Following Moghimi *et al.* (2013), an enhanced bottom roughness length  $z_0^{b_0}$  is computed as a function of the base roughness  $z_0$  and wave properties (e.g., the bottom orbital velocity of the waves) according to Styles and Glenn (2000). This allows accounting for the generated turbulence at the bottom due to the nonresolved oscillating wave motion. In the two-way coupling experiments, the GETM model provides the water level and ambient current to WAM.

The coupling between GETM and WAM is performed via the coupler OASIS3-MCT: Ocean, Atmosphere, Sea, Ice, and Soil model at the European Centre for Research and Advanced Training in Scientific Computation Software (Valcke *et al.*, 2013). The name OASIS3-MCT is a combination of OASIS3 (Ocean, Atmosphere, Sea, Ice, and Soil model coupler version 3) at the European Centre for Research and Advanced Training in Scientific Computation (CERFACS) and MCT (Model Coupling Toolkit), which was developed by Argonne National Laboratory in the USA. The details of the properties and use of OASIS3 can be found in Valcke (2013). The exchange time between models is

five minutes. This small coupling time step is a major advantage for modelling fastmoving storms compared to off-line (without using a coupler) coupled models, as in Staneva *et al.*, (2016), where hourly wave fields are used in GETM.

#### 5 **2.4 Study period (meteorological conditions)**

This study is focused on the period during the winter storm Xaver that occurred on the 5th and 6th of December, 2013, and caused flooding and serious damage to the southern North Sea coastal areas. During 4<sup>th</sup> to 7<sup>th</sup> of December, the storm depression Xaver moved from the south of Iceland over the Faroe Islands to Norway and southern Sweden and further over the Baltic Sea to Lithuania, Latvia and Estonia. It reached its lowest sea level pressure on the 5<sup>th</sup> of December at 18 UTC over Norway (approximately 970 hPa, Fig. 2 and 3). Over the German Bight, the arrival of Xaver coincided with high tides; therefore, an extreme weather warning was given to the coastal areas of north-western Germany due to high tides and wind gusts of greater than 130 km/h (Deutschländer *et al.*, 2013). The extremely high water level and waves triggered sand-displacement on the barrier islands and erosion of dunes in the Wadden Sea region. The German Weather Service reported the storm to be worse or similar to the North Sea flood of 1962, in which 340 people lost their lives in Hamburg, saying that improvements in sea defences since that time would withstand the storm surge (Deutschländer *et al.*, 2013, Lamb and Frydendahl, 1991).

20

10

15

#### **2.5 Numerical experiments**

For the control simulation (CTRL run), GETM is run as a fully three-dimensional baroclinic model without coupling with the wave model. The effects of using different coupling methods are studied by comparing the two-way fully coupled GETM-WAM model simulation (FULL run) with the one-way coupled model, in which the circulation model obtains information from WAM (one-way coupling). We denote this experiment FORCED run. Additionally, we run the circulation model GETM as a two-dimensional barotropic model (2-D run). In the final experiment, we exclude the river runoff forcing (NORIV run). The list of experiments is given in Table 1.

30

#### **3.** Observational data

The tide gauge observations from the eSurge project (www.esurge.org) are used. An overview of the existing operational tide gauges in the North Sea and Baltic Sea regions are available at the webpages of the EuroGOOS regions NOOS (North West Shelf Operational Oceanographic System) and BOOS (Baltic Operational Oceanographic System), respectively, www.noos.cc and www.boos.org. The water level data are acquired through the NOOS ftp server.

The in situ wave data are taken from the wave-buoy observational network operated in the North and Baltic Seas by the BSH (http://www.bsh.de/de/Meeresdaten/Beobachtungen).

Additionally, for validation, we use satellite measurements of the significant wave height and sea 10 level in the German Bight derived from the Jason-2, CryoSat-2 and SARAL/AltiKa altimetry satellite missions. This is of special interest since the satellite passed over the North Sea during Xaver. As explained in Fenoglio-Marc et al. (2015), the standard altimeter products are extracted from the Radar Altimeter Database System (RADS) (Scharroo, 2013). The sea water level corresponding to the instantaneous in situ tide gauge measurement, which was called Total 15 Water Level Envelope (TWLE) in Fenoglio-Marc et al. (2015), is estimated as the difference between the orbital altitude above the mean sea surface model DTU10 and the radar range corrected for the ionospheric and tropospheric path delay, solid Earth, sea state bias and load tide effects. Corrections for the ocean tide, the atmospheric inverse barometer effect and wind are not used. The storm surge is estimated by correcting the TWLE for the ocean tide given by the global ocean tide model GOT4.8 (Ray et al., 2011), see Fenoglio-Marc et al. (2015) for more details.

## 4. Model validation during extreme storm surges

#### 25 4.1 Wave model performance

In this section, we analyse the wave model performance during Xaver using the FULL experiment. The significant wave heights  $(H_s)$  from the model simulations are in good agreement with the measured values. As can been seen in the time-series graph for Elbe (top) and Westerland (bottom) stations, the measured  $H_s$  was greater than 7.5 m during 2-8 of December, 2013 (Fig. 4). The peak of Hs during the storm is reached earlier in the model simulations compared to the observations (Fig. 4b, d). This could be due to the DWD wind data (see also

5

20

Wahle *et al.*, 2016). In addition, the maximum of statistical wave height simulated by the model for the two locations (Fig. 4a, c) occurs earlier than that of the measurements, which is due to the shifted maximum of the DWD wind forecasts. The standard deviation between the model and the measurements is 0.16 m for Elbe and 0.12 m for Westerland station. The correlation coefficients between the WAM simulations and measurements are greater than 0.9 for all stations, and the normalized RMS error is relatively low (between 0.09 and 0.16 m). For the analyses of the wave model performance, including different statistical parameters computed during the extreme event for all available German Bight stations, we refer to Staneva *et al.* (2016).

10

5

The wave spectra at the FINO-1 and Elbe BSH buoy stations are given in Fig. 5 for the study period. The wave spectra from the model simulations (Fig. 5a, c) are in a good agreement with the spectra from the observations (Fig. 5a, c). The time variability of the spectral energy is accurately reproduced by the model, and the energy around the peak is similar in the observations and simulations; however, the model patterns are smoother than the observed patterns.

15 In addition to the in situ measurements, the satellite altimetry data provide a unique opportunity to evaluate both the temporal and spatial variability simulated in the model along its groundtrack at the time of the overflight of the German Bight, lasting approximately 38 sec (see Fig. 6a, b). The modelled Hs varies along the satellite ground-tracks between 1.2 and 1.9 m during calm conditions on 3<sup>th</sup> of December, 2013 at 18:00 UTC (Fig. 6a), while during Xaver, Hs varies between 6.3 m and 9.4 m (6<sup>th</sup> of December 2013 at 04:00 UTC, Fig. 6b). The spatial distribution 20 of Hs (Fig. 6c, d) is in good agreement with the satellite data in both cases. The latitudinal distribution of *Hs* simulated by the wave model (green dots) is smoother than that of the satellite data. This can be explained by the different post-processing of the satellite data of the significant wave height and by the statistical nature of its estimate by the model. For calm conditions (Fig. 6c), Hs is slightly underestimated (approximately 15 cm) in the coastal area and overestimated 25 (approximately 20 cm) in the open German Bight. During Xaver, the model slightly overestimates the satellite data in the open areas (20-30 cm). These results are consistent with the results of Fenoglio-Marc et al. (2015), who compared the SARAL data with the DWD wave simulations.

30

#### 4.2 Sea level and wave-induced forcing

In this section, we demonstrate the performance of the hydrodynamic model to simulate the mean sea level and present statistics obtained for the study period. Detailed statistical analyses of the model comparisons with measurements for the area of German Bight are quantified by Staneva et al. (2016), where the coupled model performance is shown to be in a good agreement with observations, not only during the calm conditions but during storm events. Therefore, we only provide new examples of model-data validations, including satellite data that have not been used in previous studies.

10

5

The geographic representation of the bias between the model simulations and all available tide gauge data shows that the bias for most tide gauge stations is within  $\pm -0.1$  m (Fig. 7). Exceptions are found in some coastal tide gauge data stations in the very shallow areas. This can be attributed to the relatively coarse spatial resolution (1 km) and smoother model bathymetry in the shallow coastal waters. Storm surges are estimated by subtracting from the simulations and tide gauge observations the ocean tide estimated using the T TIDE routine (Pawlowicz et al., 2002). The comparisons between individual simulations are only marginally affected by tidal 15 simulation errors because the simulations share the same systematic tidal errors. Estimating the surge component, the direct influence of tidal simulation errors in over-tides is minimized because the surge signals from observations and model runs are derived by subtracting an individual estimate of the tidal signal for each dataset.

From the comparison between the surge model and satellite data (Fig. 8), it can be concluded that the model results are in good agreement with the observations. This holds for calm conditions 20 (3<sup>th</sup> of December 2013), when the surge was weak (less than 10 cm offshore and up to 25 cm near the coastal area, Fig. 8c), as well as during Xaver on 6<sup>th</sup> of December 2013, when the surge reached almost 3 m. The statistics from the comparisons between the observations and experiments are presented in Table 2. The coupling between circulation and waves significantly 25 improves the surge predictions; when the effects of the interactions with waves are considered, both the bias and the RMSE are substantially reduced (see Table 2).

The temporal evolution of the water level for the Helgoland tide gauge data (see Fig. 1 for its location) is shown in Fig. 9. The consistency between the model simulations from the CTRL and FULL runs is very good during normal meteorological conditions; however, during the storm, the water level simulated by the stand-alone circulation model is approximately 30 cm lower than the data from the Helgoland tidal gauge station. When the wave-induced processes are considered, the simulated sea level (FULL run) approaches the observations. Including wave-

current interactions improves the root mean square error and the correlation coefficient between the tide gauges data and the simulated sea level over the German Bight (Table 2).



The surge height reaches approximately 2.5 m during Xaver, with its maximum at low water. During Xaver, two surge maxima ( $S_{max1}$  and  $S_{max2}$  in green line Fig. 9) are observed. Fenoglio et al. (2015) described the first surge maximum as a wind-induced maximum. They found that at Aberdeen and Lowestoft stations, the surge derived from the tide gauge records had only one maximum, reaching the eastern North Sea coastal areas (anticlockwise propagation) approximately ten hours later than Lowestoft (easternmost UK coast), causing the second storm surge maximum detected by the measurements in the German Bight. As shown by Staneva et al. (2016), the wave-induced mechanisms contribute to a persistent increase of the surge after the first maximum (with slight overestimation after the second peak). At the two maxima, the observed water level at the Helgoland tide gauge is in better agreement with the coupled model (FULL run: black line) than the CTRL simulated water level. The two maxima are underestimated by the stand-alone circulation model (CTRL: red line), especially at high water, when the surge difference between the model results and the measurements is approximately 30 cm for the first peak and more than 40 cm for the second peak (Fig. 9).

#### **5. Process studies**

#### 20 5.1 Sensitivity of surge predictions to coupling with waves

In this section, we analyse the role of wave-current interactions in the storm surge model and demonstrate the sensitivity to one-way versus two-way coupling. Fig. 10 shows the time series of the water level (black line) and the storm surge (red line) for six stations (see Fig. 1 for their locations) together with the differences in the surge between the FULL and CTRL runs (FULL-CTRL: green line) and the differences between the FULL and FORCED runs (FULL-FORCED: blue line). The surge during the extreme exceeds 2 m in the open-ocean stations and increases to 2.8 m near the coastal stations. The two storm surge maxima during Xaver (described in Section 4) are seen at the near-coastal station ST1-4, whereas at ST6 (in the Elbe Estuary), the surge remains at high, even in the period between the two maxima. The coupling with waves leads to a persistent increase in the surge, especially after the occurrence of the first maximum  $(S_{maxl})$ . The difference in the simulated surge between the FULL and CTRL runs (green line) reaches a maximum during the first peak of the surge and is substantial during the following two days. For



the Hörnum station (ST3), the increase in the surge due to coupling with waves exceeds 35% compared to the CTRL data (Fig. 10c). At the north-easternmost station (ST4), the surge difference between the FULL and CTRL runs is greater than 70 cm, which results in a contribution of the wave-current interaction processes of greater than 40%. For the deeper openwater station (ST5, Fig. 10f), the maximum contribution is approximately 30 cm, a 25% increase in the surge. The differences between the FORCED and FULL runs are relatively small (less than 4% of the total for all stations, see the blue line). However, for the shallower Elbe Station (ST6, Fig. 10e), the effects of two-way coupling compared to the FORCED run (one-way coupling) are important. Staneva et al. (2016) provided a summary of improved model performance with respect to the prediction of the sea level, which is the main variable considered below in the analysis of extreme surges in the German Bight. The quantification of the performance shows that in a large number of coastal locations, both the RMS difference and the bias between the model estimates and observations are significantly reduced because of the improved representation of physics. Only in very few very near-costal tide gauge stations does the coupling not lead to improvements, which might be due to the insufficient resolution of the near-coastal processes in very shallow water regions.

To provide an illustration of the coastal impact caused by Xaver, we analyse the horizontal patterns of the maximum storm surge (Fig. 11) over the four tidal periods T1-T4 (as specified in Fig 9). During the second peak (T3), the surge exceeds 2.8 m over the whole German Bight coast (Fig. 11c); the storm surge near Elbe is greater than 3 m. During the period of the first surge peak (T2, Fig. 11b), the maximum occurs in the Sylt-Römo Bight area (above 2.8 m) and along the Elbe and Weser estuaries (approximately 2.5 m). Over the whole German Bight, the simulated surge exceeds 1.5 m. In the period of relatively calm conditions before the storm (T1), the surge is relatively low (Fig. 11a, less than 30 cm). A decrease in the surge towards the north-western German Bight is simulated during T4 (Fig. 11d). The intensification of the storm surge from the open sea towards the coastal area is consistent with the specific atmospheric conditions during Xaver (Fig. 3).

15

25

30

To better understand the impact of wave-current interactions on the surge simulations, we also analyse the horizontal patterns of the maximum differences in the storm surge between the coupled model (FULL run) and the stand-alone GETM (CTRL run). The maximum differences for each grid point are estimated over the four tidal periods (Fig. 12, T1-T4). The patterns show that the differences between the FULL and CTRL runs during the first surge maximum are more

noticeable in the very shallow North Frisian Wadden Sea. The maximum surge simulated by the fully coupled model exceeds that of the CTRL run by approximately 60 cm along the Sylt-Römo Bight during T2. The enhancement of the surge in the coastal area (see Fig. 11b) may be due to the nonlinear interaction between circulation and waves (the contribution of the wave-current interaction to the increase of the surge is greater than 25%) along the German Bight coastal region (Fig. 12a). For T3, the maximum surge difference (approximately 55 cm) is concentrated along the Elbe River; however, the increase in the surge due to wave-induced processes exceeds 40 cm along the entire German Bight coast. During the second Xaver peak, the radiation stress contributes to a rise in the sea level, which is directed towards the Elbe-Weser river area. During the first peak (T1), the differences between FULL and CTRL are more pronounced near the North Frisian Wadden Sea. The computed maximum surge differences are higher during T2 than during T3. For T4 (Fig. 12d), the maximum difference of approximately 15 cm occurs for the east Frisian coast towards Elbe River area, whereas in the north-eastern area, the wave-induced processes do not contribute much to the mean sea level and the surge simulations of the FULL runs are similar to the CTRL run. The horizontal distribution of the patterns of Fig. 12 demonstrates the good consistency with the meteorological situation (Fig. 3). The effects of wave-induced forcing during the storm are also noticeable in the open North Sea (maximum surge differences are approximately 30 cm Fig. 12b, c) due to the dominant role of the radiation stress—even in the deeper areas, the differences between the FULL and CTRL surge estimates are greater than 20%. Although the wave heights are much higher in the open sea, the water there is much deeper; thus, the differences in sea level between the FULL and CTRL runs are relatively small.

#### 5.2 3-D versus 2-D barotropic models

25 Depth-averaged two-dimensional flow models are widely applied in storm surge simulation and have been assumed to meet the requirements of operational forecasts. They are also widely used in many scientific studies. However, to study the flow characteristics of storm surges, the use of only barotropic models is insufficient, especially in large estuaries. The flows in the surface and bottom layers are usually quite different, so depth-averaged two-dimensional models cannot sufficiently depict the flow structure. Furthermore, storm surge models do not account for 30

baroclinic processes, such as density-driven changes in water masses, which are important in estuarine environments.

The changes in the sea level due to temperature for the Nederland coastal areas have been studied by Tsimplis et al. (2006). Dangendorf et al. (2013) showed that laterally forced steric variation and baroclinic processes are important at decadal scales, while atmospheric forcing causes the annual variability in the sea level. Chen et al. (2014) studied the role of remote baroclinic and local steric effects in the interannual sea level variability and found that a threedimensional model that considers the temperature and salinity can more accurately simulate the changes in the water level related to the North Atlantic Oscillation (NAO). In these models, more realistic open boundary conditions (than in the barotropic models) are used to account for the dynamics of heat and salt. We quantify the benefit of using a fully three-dimensional model that also considers temperature and salinity to simulate the sea level during extremes.

The surge differences between the FULL and 2-D runs are much larger during Xaver (T2, Fig. 13b) than during calm conditions (T1, Fig. 13a). For T2, the maximum difference increases 15 eastward from 2-5 cm at the western boundary of the German Bight to more than 80 cm along the North Frisian Wadden Sea coast and near the Elbe and Weser estuaries. The surge differences decrease to 30 cm during the second peak of Xaver. After the storm, the threedimensional effects contribute to an increase in the sea level in the direction of the Elbe Estuary (Fig. 13d). These effects can exceed 25% of the sea level increase, compared to the 2-D model 20 simulations (Fig. 14). For the Elbe area, the 2-D model underestimates the mean sea level by approximately 1 m. This could cause significant underestimation of the sea level predictions of the barotropic models. For T4, the impact of baroclinicity is localized along the south-eastern coastline (Fig. 13d). The differences between FULL and NORIV (Fig. 14, blue line) are negligible at ST3, while at ST6, they are approximately 15 cm in the vicinity of the Elbe Estuary 25 during the storm Xaver. When analysing the impact of the baroclinicity on sea level model results, we use the barotropic hydrodynamic model that is not coupled to the wave model since our aim is to demonstrate only those effects. Introducing wave-circulation coupled processes (as demonstrated in the previous sections) to the barotropic model can reduce the differences between this model and the FULL run.

5

10

#### 6. Discussion and conclusions

With the uncertainties of storm surge predictions under climate change, the quantification of associated hazards is of great interest to coastal areas. The demand to understand the risk of damage has increased for the development of future climate scenarios.

The accurate real-time assessment of storm surges and inundation areas is unable to fully satisfy these demands because atmospheric storm forecasting, as the important driving force of surges, is not perfect. This leads to a high degree of uncertainty in storm surge forecasting. The peak surge depends on the accurate prediction of the landfall position and time. The future development of water level predictions will focus on enlarging the observation data network and further model developments. To reduce uncertainty, increasing knowledge of various processes, such as tide-wave-surge interactions, is needed. Improved weather forecasts and further coupling between the atmosphere, ocean and wave components will reduce the uncertainty. Increasing the horizontal resolution in the near-coastal areas is made possible by the availability of more computational resources. In this study, we show that the wave-dependent approach yields a 25% larger surge over the whole coastal area of the German Bight, reaching a contribution of approximately 40% in some coastal areas during extremes. The fully 3-D model and the barotropic model produce approximately 20% difference in the water level of the coastal areas of the German Bight during Xaver. The possible advantages of including the wave-current interaction in two dimensional barotropic models to improve sea level predictions will be the subject of further studies.

We demonstrated that the consistency between the observations and model simulations of the circulation model only and the coupled-wave circulation model is good during normal meteorological conditions. However, during the storm event, the water level simulated by the stand-alone circulation model is approximately 30 cm lower than the observations. When the wave-induced processes are considered, the simulated sea level (FULL run) is closer to the observations, and the statistics between the tide gauge data and the simulated sea level over the German Bight are improved. Wave-induced mechanisms contribute to a persistent increase in the surge after the first maximum (with slight overestimation after the second peak) during Xaver. The two maxima are underestimated by the stand-alone circulation model, especially at high water, where the surge difference between the model results and the measurements is

15

approximately 30 cm for the first peak and more than 40 cm for the second peak. When estimating the surge residuals, the direct influence of tidal simulation error is minimized because the surge signals from the observations and models are derived separately by subtracting an estimate of the tidal signal for each dataset

- 5 New observations have recently become available from remote sensing of wind speed, waves, sea levels and currents using X-band and HF-radar, ADCP, LIDAR, Ku and Ka band pulselimited and delay Doppler radar altimetry, which promise high-quality space observations in the coastal zones. Better sea level data near landfall and storm variables are provided by an improved network of tide gauges and buoys and observations from space. According to the 10 balance of investment and the demand of disaster relief, more tide gauge stations should be established in empty or sparse areas. These newly available remote sensing data are expected to improve forecasting model systems (both ocean and atmosphere). For coastal areas, the role of wave-induced forcing on coastal morphology should also be the subject of further study.
- For regions such as the German Bight, the role and potential uncertainties of the shallow water 15 terms in the wave model are also of great importance since shallow water regions with the strongest wave-ocean interactions are contributed by these terms during extreme storm surge events. The shallow water terms in the action balance equation increase rapidly with decreasing depth. Depth and current refraction, bottom friction and wave breaking play dominant roles in very shallow water during storm events. The wave breaking term prevents unrealistic high waves 20 in such situations near the coast. Since the wave model results are representative of a model grid cell, the shallow water terms involve uncertainties due to the choice of a realistic bathymetry and the spatial resolution of the model grid.

Wahle et al. (2016) studied the effects of coupling between an atmospheric model and a windwave model and found a reduction of surface wind speeds and a reduction of simulated wave heights. Their results revealed that the effect of coupling resulted in significant changes in both wind and waves and that the two-way coupling between the atmosphere and wave models further improved the agreement between observations and simulations. Our modelling system will be extended by integrating the latest developments in atmosphere-wave-current interactions towards a fully three-way coupled system to further investigate the effects of coupling on storm surges.

25

A rise in the sea level combined with high waves can increase the intensity of coastal flooding, causing a collapse of and damage to seawalls and levees. Improved wave and ocean circulation forecasts for the North Sea and its coastal areas, especially the German Bight, are of great importance for the marine and coastal environment since early warnings and protection can contribute to reducing the damage caused by flooding and coastal erosion. This is of utmost importance for offshore wind energy farms, ship routing, and coastal zone protection.

We demonstrated that the interaction between waves and three-dimensional hydrodynamic models reduces forecast errors, especially during extreme events. This will enable further use of high-resolution coupled models to improve coastal flooding prediction and climate studies.

#### 10

5

#### Acknowledgements:

This work has been supported by the Coastal Observing System for Northern and Arctic Seas (COSYNA) and as part of the Copernicus Marine Environment Monitoring Service (CMEMS) Wave2Nemo project. CMEMS is implemented by Mercator Ocean in the framework of a delegation agreement with the European Union. Luciana Fenoglio is supported by the European Space Agency (ESA) within the Climate Change Initiative (CCI). The authors are grateful for I. Nöhren for assistance with the graphics and BSH for providing the observational data. The authors thank Sebastian Grayek for assistance with the tidal model.

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

| Experiment | 3-D GETM | WAM     | Barotropic | <b>Rivers run-off</b> |
|------------|----------|---------|------------|-----------------------|
| CTRL       | yes      | -       | -          | yes                   |
| FULL       | yes      | Two-way | -          | yes                   |
| FORCED     | yes      | One-way | -          | yes                   |
| 2D         | -        | -       | yes        | no                    |
| NORIV      | yes      | Two-way | -          | no                    |

 Table 1. Model experiments.

5

|             | CTRL  | FULL  | FORCED | 2D    |
|-------------|-------|-------|--------|-------|
| RMSE        | 0.26  | 0.16  | 0.15   | 0.39  |
| Bias        | -0.17 | -0.09 | -0.10  | -0.28 |
| Correlation | 0.84  | 0.92  | 0.93   | 0.76  |

**Table 2.** Surge (m): Root-Mean Square Errors (RMSE), bias (model-observations) and correlationbetween storm surge component from four model runs (CTRL, FULL, FORCED and 2D) and from tidegauge records of the British Oceanographic Data Centre (BODC) over the German Bight area.

Figures