# Peer review of "Coastal flooding: impact of waves on storm surge during extremes. A case study for the German Bight"

_Natural Hazards and Earth System Sciences, 2016_

## Referee Comment (RC1) · Anonymous Referee #1 · 22 Aug 2016

—- General Comments

As a detailed assessment of a coupled high resolution wave-ocean modelling system's sensitivities in an extreme case, this paper provides a useful addition to existing published evidence regarding coupled systems and is a valid extension of the work in Staneva et al, 2016. I would therefore recommend this paper for publication, but with some additions/corrections related to the points below.

—- Specific Comments

Section 2.1 Whilst this information may well be published in the authors' previous papers, it would be useful to those reading this paper in isolation if some extra details on

the update frequencies of atmosphere and river forcing data were provided.

Section 2.2 This is an extreme case in shallow water, so please could the source term parameterizations for bottom friction and depth induced breaking dissipation that were used in the wave model be stated?

Section 2.3 I found the statements that " is the sum of the Eulerian current and the Stokes drift" and "Thus the divergence of the radiation stress is the only (to second order) force related to waves in the momentum equations." somewhat contradictory. In the equations, Mellor (2011) has been followed correctly and I see the basic point about radiation stress being the difference between coupled and uncoupled systems, so just wondering if the authors can review the text in this section for clarity.

Section 2.5 As per the comment for section 2.1, can the frequency of coupling fields exchange be added please? Also, please note in Table 1 whether the NORIV wave model is one or two way coupled for consistency with the rest of the table.

Section 4.1 It's not clear whether the wave model discussed in this section and associated figures is the two way coupled version or the stand alone wave model. Can this be made more explicit?

Section 4.1, p9, line 9, Looking at the figure I get the impression that the peak of the storm is simply mis-timed rather than over predicted, unless the authors are discounting the measured peak for some reason. Please check.

Section 4.2 and later discussions. If I understand this correctly, the surge residual is defined by subtracting the same predicted tidal residual (generated via the T_TIDE package) from observations and model alike. The model residual is therefore a combination of both the model error in background tide prediction plus error in the surge prediction. In that case, I think it is important that any known systematic error in the model tide is stated in order to contextualise the benefits of the wave coupling. If these errors are not well understood, then I would recommend that the potential errors associated with the model tide are acknowledged and caveated in the discussion.

Section 5.1 Please comment on whether the coupling improved results at all individual stations, or just most of them...

Section 5.2 Regards the comparison with the barotropic model:

1) One of the arguments presented by the authors relates to large scale inter-annual effects on background water level, which a barotropic model will not deal with; this is correct, but can be mitigated to some extent if the predictive system for water level comprises an astronomic prediction of water level based on observations (which will include these long term effects) plus the barotropic model's estimate of the surge residual - this approach is adopted operationally in the UK for example. In terms of this paper one question for the authors to address is whether they believe that these effects are not present in the T_TIDE data used to calculate the residuals they show?

2) Of more importance, the barotropic model presented does not include any barotropic coupled effects (which might be included due to both waves radiation stresses and also water volumes associated with river inputs?) - however, the text implies that the main difference is baroclinicity. In order to make this argument better it would be good if the authors could present why they believe that introducing some coupled processes to the barotropic model would not close the gap between this model and the FULL run?

3) Finally, please check Figure 14, where the surge line for station ST3 does not look consistent with that in Figure 10.

Section 6 Items to consider for addition to the discussion:

1) the wave model, via the atmosphere model I expect, has over-predicted during period T2 and then been about right for period T3. In support of the comments regarding atmospheric uncertainties, how did the comparisons of modelled and observed surge vary during these periods for the FULL run?

2) in these simulations, there is no feedback to the atmospheric model from the waves,

so the coupled system is not fully closed. In terms of the argument being presented here, where the waves are strongly affecting the ocean model in a shallow water region, I'd imagine that the sensitivity to the atmosphere-wave-ocean coupling is not too big a consideration at these scales; however, it might be useful to acknowledge this point more than has been presently done on line 9,p15.

3) Is it possible for the authors to discuss/speculate further on the role and potential uncertainties of the shallow water terms in the wave model?. My impression in this case is that the region with strongest wave-ocean interactions will see strong contributions from these terms in such a large storm and shallow depths.

—- Technical Comments/Proposed text corrections

Page 2, para 2 There are a number of typos and the grammar could be improved significantly in this paragraph and, if kept, the authors need to review this carefully. However, in the context of the paper I think that the arguments being made about climate changes effects and other reasons for improving model accuracy can be taken as read (or just briefly expanded upon in the first paragraph); so I'd suggest removing this paragraph altogether.

p2, line 2, "predictions of the sea" -> "predictions of sea"

p2, line 6, "demand of improving" -> "demand for improving"

p2, line 8, "role at shallow area" -> "role in enhancing sea-surface elevation in shallow water areas"

p2, line 12 "mixing to circulation model." -> "mixing to a circulation model."

p2, line 17 "of biogeochemical" -> "biogeochemical"

p2, line 18, "radiation stress that accounts" -> "radiation stress accounts"

p2, line 20, "by number of studies like" -> "by a number of studies, such as"

p2, line 30, "distributions s" -> "distributions is"

p3, line 30, "area are substantial hazard" -> "area are a substantial hazard,"

p3, line 31, "The coastal" -> "Coastal"

p4, line 22, "in details" -> "in detail"

p4, line 24, "are-sea" -> "air-sea"

p6, line 10, "equation of motions" -> "equation of motion"

p8, line 1, "The effects on using" -> "The effects of using"

p8, line 6, "rive" -> "river"

p8, line 11, "regions available" -> "regions are available"

p10, line 4 "in good consistency" -> "are consistent"

p10, line 10, "analyses on model" -> "analyses of model"

p10, line 13, "new examples on" -> "new examples of"

p10, line 28, "on Tables 2" -> "in Table 2"

p11, line 14 "reached reaching" -> "reached"

p12, line 16, "comparissons" -> "comparisons"

p13, line 11, "in direction to" -> "directed toward"

p13, line 31, "Even more" -> "Furthermore"

p14, line 7, "the three dimensional model" -> "three dimensional models"

p14, line 30, "hazard is" -> "hazards are"

p15, line 1, "has been gradually mature" -> "has gradually matured"

p15, line 2, "defective to satisfy the" -> "unable to fully satisfy these"

p15, line 3, "the real time storm forecasting" -> "atmospheric storm forecasting"

p15, line 3, "is not perfect in practical use." -> "is not perfect."

p15, line 4, "It always" -> "This"

p15, line 5, "is depended" -> "depends"

p15, line 5, "accurate landfall position" -> "accurate prediction of landfall position"

p15, line 6, "tide may has a huge tide range" -> "tide may have a huge range"

p15, line 7, "forecasting cause" -> "forecasting can cause"

p15, line 9, "increasing the knowledge on" -> "increasing knowledge of"

p15, line 10, "weather forecast" -> "weather forecasts"

p15, line 13, "that wave-dependent approach yields to 25% larger surge at" -> "that the wave-dependent approach yields a 25% larger surge over"

p15, line 14, "German Bight reaching a contribution of about 40% is some coastal area" -> "German Bight and reaching a contribution of about 40% in some coastal areas"

p15, line 15, "The contribution of the fully 3-D model in comparison with a 2D barotropic one yield to" -> "The contribution of a fully coupled 3-D model in comparison with an uncoupled 2-D barotropic one yielded up to"

p15, line 23, "demand of disaster relief" -> "demands of disaster relief"

---

## Referee Comment (RC2) · Anonymous Referee #2 · 30 Aug 2016

In the presented manuscript, the authors further investigate the data and results of their previously published paper "Coupling of wave and circulation models in coastal–ocean predicting systems: a case study for the German Bight, Ocean Sci., 12, 797-806, doi:10.5194/os-12-797-2016, 2016 by Staneva et al.

In the new manuscript, more emphasis is given on the storm surge predictions from coupled -and uncoupled models, while the general experiment setup and case study are the same as in the previous paper. The advantage of having a coupled model system is discussed; the authors show that the coupling of their ocean circulation model with a wave prediction model improves the predictions of extreme storm surges to a large degree.

[Figure]

While little new data or knowledge is presented compared the the previous paper, I would still consider this new paper as worthy for publication because their model data shows remarkable agreement with observations, particularly when using a coupled model system. The relevance of wave-current interactions for storm surges still lacks sufficient documentation that is backed up with observational data, and this new paper presents strong arguments for using coupled models for the forecasting of dangerous storm surges. The data is presented clearly and informative in the figures, but the text needs some revision with regard to clarity and English grammar, therefore I would recommend the paper for acceptance with minor revisions.

Points to be corrected:

- Some references that are used in the text are missing in the the reference list.

- page 2, line 8: wind-induced surface stress does generally play an important role, not only in shallow areas.

- page 2, line 11: The reference to Qiao et al (2004) is not an original reference to this problems, there are many earlier studies that treat wave-induced mixing in both experiments and models. It would be good to also cite some of the earlier works here.

- page 6, line 12: If  is the sum of Eulerian current and the Stokes drift, equation (3) will solve for the Lagrangian current following water masses. This is somehow different to the way GETM solves for fixed grid points. If solving for  that includes Stokes drift, the radiation stress is not the only wave information that is used in eq. (3). Note that traditional formulations of radiation stress use a Eulerian framework. I think that that , as it is used here, should only include the Eulerian current. If not, further revision of the coupling method will be necessary.

- The coupling from GETM to to WAM should also be described along with section 2.3.

- Some text passages, particularly section 6 are somehow hard to read and should be revised for clarity and grammar.

---

## Referee Comment (RC3) · Anonymous Referee #3 · 13 Sep 2016

The manuscript presents a case study analysis of coastal inundation during an extreme extratropical storm event (Xaver) that made its land fall in northwest Europe December 2013), with a considerable impact in the North Sea. The central focus (and goal) of the paper is to show the goodness of having a wave model coupled to a surge model (or a regional ocean model).

I consider this study useful and interesting, nevertheless I have some comments regarding the way the study and the results are presented. I make some suggestions regarding language, but the authors should read and correct the whole text, since the use of the English language is sometimes far from appropriate. If the authors choose to comply with my comments I would still like to read the manuscript before the final

acceptance.

Abstract The abstract has several flaws that I suggest the authors should address. Please have in mind that the abstract should "survive" (or stand) by itself. Hence it should have concise but complete information so that an educated reader knows (or understands) what to expect in the text body. Please provide information about the models you are using in the abstract.

P1-L13 (same in L21 and L23): Extremes? What extremes? Extreme storm event? Extreme sea level rise? P1-L21: replace "enhances significantly" with "is significantly enhanced". P1-L23: replace "area" with "areas". P2-L7: erase "the" before "ocean". P2-L7: Regarding sentence starting with "The wind-induced..." why is this here? It seams disconnected from the rest of the text (although, of course, being a valuable statement). P2-L8: sea surface or ocean surface (mixed) layer? I tend to look at the sea (ocean) surface as a skin layer. Please be clearer. P2-L13: add "a" before "circulation". P2-L15: replace "waves-current" with "wave-currents" (here and in other parts of the text). P2-L16: I am afraid wave models are not earth system components. Regarding ". . . and further integrating of biogeochemical or morphologic parts" I don't get what you mean; could you please re-phrase it? P2-L20: instead of "wind boundary layer" (which doesn't exist or it is not a valuable geophysical statement" please use "lower marine atmospheric boundary layer". All references here are from high wind speed regimes, when the highest (deeper) impact actually occurs during light winds and swell regimes. Consider adding some references regarding light winds regime. P2-L22 (and in several other parts of the text): add curly brackets on the years in the references. P2-L26: what do you mean with "radiation stress approach"? P2-L27: what is a "practical analysis". I am afraid this might not be a very scientific statement. P2-L28: what is "circulation for the ocean state"? P2-L30: the sentence starting with "The role of. . ." is lost here. No relation with before of after text. P3-L1: add "a" before "Lagrangian". "Drift" what drift? Stokes? Wave induced? P3-L3: replace semicolon with full stop and start new sentence afterwards. No need for this here (here and in other parts of

the text). P3-L7: it is a fact that storm surges are meteorologically driven, not a "well accepted" situation. It would be the same as saying that "it is well accepted that ocean surface gravity waves are wind driven", or that "the thermohaline circulation is driven by water density differences". P3-L11: correct tense of sentence starting with "IPCC...". P3-15: please provide some more explanation on how waves and tides are amplified by the rise of sea level. P3-L16: "could" or "can"? P3-L17: add "and" after "seawalls"; add "ocean" before "circulation". P3-L18: add a comma after "Bight"; replace "greatest" with "great". P3-L19: how can the forecast reduce the damage? P3-L20: add "farms" after "energy"; replace "navigation" with "routing". P3-L23: sentence starting with "Further..." is confusing; please re-write. P3-28: what are external waves? P3-L30: replace "substantial" with "a considerable"; replace "for" with "in". P4-L1: erase "cause". P4-L8: erase "as well as satellite data"; add "and remote sensing" after "insitu". P4-L22: "outer model" or "outer domain"? P4-L28: add "further details." after "2016)". P5-L15: "action density" or "wave energy density"? P5-L21: there is no "S" in the rhs of equation (2). P6-L9: "wave motion" is too broad; please provide additional explanation. P6-L27: add "wave model" after "by"; the WAM model doesn't "give" data!; all this sentence is inaccurate from a wave model standpoint. P7-L2: add "of" before "GOTM". P7-L10: replace "causing" with "that caused". P7-L19: replace "has" with "had". P8-L19: erase double punctuation. P9-L5: replace "As an example we present" with "As can be seen in". P9-L12: sentence starting with "The standard..." is confusing; please consider re-writing. P9-L15: "low"?, how much?; replace "analysis on" with "the analysis of". P9-L28: you have defined Hs before, hence erase "significant wave height". P10-L12: replace "is" with "are". P10-L28: replace "demonstrate" with "show"; maybe this reduction should be quantifies here. P11-L14: reached or reaching? P11-L30: replace "their" with "its". P12-L3: erase "with". P12-L17: add "to be" before "important". P12-L30: correct the tense of the verb. P13-L4: "North-Frisian Wadden Sea" is this correct? P13-L5: "is due..." how do you know? P13-L6: replace "of the" with "to the". P13-L29: models only could be inappropriate..." wrong tense; please re-write. P14-L2: replace "is" with "are". P14-L3: add "the" before Nederland".

P14-L6: sentence starting with "Recently…" is confusing; consider re-writing. P14-L22: replace "with" with "to". P14-L30: replace "the coastal area" with "coastal areas". P14-L30: replace "know" with "understand". P14-L31: "risks and losses"? What do you mean?; replace "increases" with "has increased". P15-L1: sentence starting with "Although…" is confusing; consider re-writing. P15-L4: what "leads"? P15-L6: replace "has" with "have". P15-L7: replace "cause" with "causes". P15-L9: erase "the". P15-L18: sentence starting with "Nowadays…" is confusing; consider re-writing. P15-L24: replace "satellite" with "remote sensing"; which products?, please be more specific. P15-L27: add "have after "We"; add "the" after "that".

---

## Author Response (AR1)

Dear Prof. Trigo,

5 Thank you for your decision of 22 October, 2016 regarding our NHESS-2016-227 Manuscript.

As requested, we provided the revised version of our manuscript, in which we considered all comments and recommendations by the three reviewers.

10 Below are point-by-point responses of the reviewers' comments together with the marked-up manuscript version.

Regarding the use of English – the revised version of our manuscript has been carefully checked for typos and English grammar by native English speakers.

15

We are looking forward hearing from you soon.

Best regards, Joanna Staneva

**Answers of the reviewers' comments**

**Reviewer #1**

**5**

20

30

**--- General Comments**

As a detailed assessment of a coupled high resolution wave-ocean modelling system's sensitivities in an extreme case, this paper provides a useful addition to existing published evidence regarding coupled systems and is a valid extension of the work in Staneva et al, 2016. I would therefore recommend this paper for publication, but with some additions/corrections related to the points below.

10

--- Specific Comments

Section 2.1 Whilst this information may well be published in the authors' previous papers, it would be useful to those reading this paper in isolation if some extra details on the update frequencies of atmosphere 15 and river forcing data were provided.

Authors: More information about the model setup, including a description of the open boundary forcing, atmospheric forcing and river runoff, has been included in Section 2.1. Additional references were also added.

Section 2.2 This is an extreme case in shallow water, so please could the source term parameterizations for bottom friction and depth induced breaking dissipation that were used in the wave model be stated?

25 Authors: Additional information about the parameterizations used in our model setup, including more references, was provided in Section 2.2.

Section 2.3 I found the statements that " is the sum of the Eulerian current and the Stokes drift" and "Thus the divergence of the radiation stress is the only (to second order) force related to waves in the momentum equations." somewhat contradictory. In the equations, Mellor (2011) has been followed correctly and I see the basic point about radiation stress being the difference between coupled and uncoupled systems, so just wondering if the authors can review the text in this section for clarity.

Authors: We apologise for the confusion we created with this mis-formulation and completely agree with 35 this comment. As described in the text, we follow the procedure of Mellor (2011). This inconsistency was also mentioned by reviewer #2. Both suggestions are exactly what was used in our study. We corrected the text regarding the statement of  accordingly, and a clearer explanation is given in the revised manuscript.

- 40 Section 2.5 As per the comment for section 2.1, can the frequency of coupling fields exchange be added please? Authors: Additional information about the coupler, coupling fields, etc.-, including references, has also been provided in the revised manuscript
- 45 Also, please note in Table 1 whether the NORIV wave model is one or two way coupled for consistency with the rest of the table.

Authors: We agree and modified Table 1 to specify that NORIV is a two-way coupled model, making the third column consistent with the rest of the text.

Section 4.1 It's not clear whether the wave model discussed in this section and associated figures is the 5 two way coupled version or the stand alone wave model. Can this be made more explicit?

Authors: This point has been clarified in Section 4.1.

Section 4.1, p9, line 9, Looking at the figure I get the impression that the peak of the storm is simply mistimed rather than over predicted, unless the authors are discounting the measured peak for some 10 reason. Please check.

Authors: We agree with the statement that the peak of the storm is slightly mistimed rather than overpredicted, as shown in Figure 4, and this has been changed accordingly in the revised manuscript.

15

Section 4.2 and later discussions. If I understand this correctly, the surge residual is defined by subtracting the same predicted tidal residual (generated via the T\_TIDE package) from observations and model alike. The model residual is therefore a combination of both the model error in background tide prediction plus error in the surge prediction. In that case, I think it is important that any known systematic error in the

model tide is stated in order to contextualise the benefits of the wave coupling. If these errors are not well 20 understood, then I would recommend that the potential errors associated with the model tide are acknowledged and caveated in the discussion.

Authors: We agree that the nonlinear interaction of the storm surge signal with the systematic error in the 25 tidal simulation may have an effect on estimating the difference between the observed and the simulated surge signal. We provided further clarification in Section 4.1 and in the discussion section.

Section 5.1 Please comment on whether the coupling improved results at all individual stations, or just most of them...

**30**

**Authors: This topic has been discussed in greater detail in Section 5.1.**

Section 5.2 Regards the comparison with the barotropic model:

- One of the arguments presented by the authors relates to large scale inter-annual effects on 1) 35 background water level, which a barotropic model will not deal with; this is correct, but can be mitigated to some extent if the predictive system for water level comprises an astronomic prediction of water level based on observations (which will include these long term effects) plus the barotropic model's estimate of the surge residual - this approach is adopted operationally in the UK for example. In terms of this paper one question for the authors to address is whether they believe that these effects are not present in the 40
- T\_TIDE data used to calculate the residuals they show?

Authors: The tidal analyses in the present study consider the bias and linear drift of the tidal signal, which for the length of analysed period, a few days, may be sufficient to fit the large-scale annual and interannual signal of the background water level. However, we agree with the reviewer that for the analysis of longer periods a more sophisticated approach is advisable.

2) Of more importance, the barotropic model presented does not include any barotropic coupled effects (which might be included due to both waves radiation stresses and also water volumes associated with river inputs?) - however, the text implies that the main difference is baroclinicity. In order to make this argument better it would be good if the authors could present why they believe that introducing some

5 coupled processes to the barotropic model would not close the gap between this model and the FULL run?

Authors: Yes, when analysing the role of baroclinicity, we used the barotropic model that was not coupled to the wave model. The aim of our sensitivity studies was to demonstrate the individual effects of coupling with waves and baroclinicity separately. We agree that to some extent the introduction of coupled processes of the barotropic model would partially reduce the gap between this model and the FULL run, which is discussed in Section 5.2. The possible advantages of including the wave-current interactions in the 2-D models to improve the sea level predictions were also addressed in the discussion.

3) Finally, please check Figure 14, where the surge line for station ST3 does not look consistent withthat in Figure 10.

Authors: We apologise for the incorrect Figure 14a and thank you for noticing the error. In the revised manuscript, the correct Figure 14a has been included.

20 Section 6 Items to consider for addition to the discussion:

10

1) the wave model, via the atmosphere model I expect, has over-predicted during period T2 and then been about right for period T3. In support of the comments regarding atmospheric uncertainties, how did the comparisons of modelled and observed surge vary during these periods for the FULL run?

25 Authors: We agree with the suggestion and added comments on this issue in an additional paragraph in the discussion section.

2) in these simulations, there is no feedback to the atmospheric model from the waves, so the coupled system is not fully closed. In terms of the argument being presented here, where the waves are strongly affecting the ocean model in a shallow water region. I'd imaging that the constitution to the atmospheric model is a shallow water region.

30 affecting the ocean model in a shallow water region, I'd imagine that the sensitivity to the atmospherewave-ocean coupling is not too big a consideration at these scales; however, it might be useful to acknowledge this point more than has been presently done on line 9,p15.

Authors: We completely agree. The atmosphere-wave (COSMO-WAM) interaction is a subject of another
 study (Wahle et al, 2016). Our aim is to study and understand the wave-current interactions (the current manuscript) and wave-atmosphere interactions separately for our coupled model system before proceeding to fully three-way atmosphere-wave-ocean interactions. The latter will be the subject of forthcoming developments and studies. We included an additional paragraph addressing this issue.

- 3) Is it possible for the authors to discuss/speculate further on the role and potential uncertainties of the shallow water terms in the wave model?
   My impression in this case is that the region with strongest wave-ocean interactions will see strong contributions from these terms in such a large storm and shallow depths.
- 45 Authors: The role and potential uncertainties of the shallow water terms in the wave model have been discussed in the final section.

--- Technical Comments/Proposed text corrections

Page 2, para 2 There are a number of typos and the grammar could be improved significantly in this paragraph and, if kept, the authors need to review this carefully.

5

10

Authors: We completely agree and carefully revised the manuscript for typos and English grammar.

However, in the context of the paper I think that the arguments being made about climate changes effects and other reasons for improving model accuracy can be taken as read (or just briefly expanded upon in the first paragraph); so I'd suggest removing this paragraph altogether.

Authors: We agree with this comment and removed this part from the Introduction.

- p2, line 2, "predictions of the sea" -> "predictions of sea"
- 15 Authors: The suggested revision has been made.

p2, line 6, "demand of improving" -> "demand for improving" Authors: The suggested revision has been made.

20 p2, line 8, "role at shallow area" -> "role in enhancing sea-surface elevation in shallow water areas" Authors: The suggested revision has been made.

p2, line 12 "mixing to circulation model." -> "mixing to a circulation model." Authors: The suggested revision has been made.

25

p2, line 17 "of biogeochemical" -> "biogeochemical" This has been re-phrased, following Reviwer#3 comment

p2, line 18, "radiation stress that accounts" -> "radiation stress accounts"
30 Authors: The suggested revision has been made.

p2, line 20, "by number of studies like" -> "by a number of studies, such as" p2, line 30, "distributions s" -> "distributions is" Authors: The suggested revision has been made.

35

p3, line 30, "area are substantial hazard" -> "area are a substantial hazard," p3, line 31, "The coastal" -> "Coastal" "Coastal" Authors: The suggested revision has been made.

40 p4, line 22, "in details" -> "in detail" p4, line 24, "are-sea" -> "air-sea" Authors: The suggested revision has been made.

p6, line 10, "equation of motions" -> "equation of motion" p8, line 1, "The effects on using" -> "The effects of using" p8, line 6, "rive" -> "river"

45 Authors: The suggested revision has been made.

p8, line 11, "regions available" -> "regions are available" p10, line 4 "in good consistency" -> "are consistent" p10, line 10, "analyses on model" -> "analyses of model" p10, line 13, "new examples on" -> "new examples of" p10, line 28, "on Tables 2" -> "in Table 2" Authors: The suggested revision has been made.

5

40

p11, line 14 "reached reaching" -> "reached" p12, line 16, "comparisons" -> "comparisons" p13, line 11, "in direction to" -> "directed toward" p13, line 31, "Even more" -> "Furthermore" Authors: The suggested revision has been made.

p14, line 7, "the three dimensional model" -> "three dimensional models" p14, line 30, "hazard is" -> 10 "hazards are" Authors: The suggested revision has been made.

p15, line 1, "has been gradually mature" -> "has gradually matured" p15, line 2, "defective to satisfy the" -> "unable to fully satisfy these" p15, line 3, "the real time storm forecasting" -> "atmospheric storm 15 forecasting" p15, line 3, "is not perfect in practical use." -> "is not perfect." Authors: The suggested revision has been made.

p15, line 4, "It always" -> "This"

20 Authors: The suggested revision has been made.

> p15, line 5, "is depended" -> "depends" Authors: The suggested revision has been made.

25 p15, line 5, "accurate landfall position" -> "accurate prediction of landfall position" p15, line 6, "tide may has a huge tide range" -> "tide may have a huge range" p15, line 7, "forecasting cause" -> "forecasting can cause"

Authors: The suggested revision has been made.

30 p15, line 9, "increasing the knowledge on" -> "increasing knowledge of" p15, line 10, "weather forecast" -> "weather forecasts" Authors: The suggested revision has been made.

p15, line 13, "that wave-dependent approach yields to 25% larger surge at" -> "that the wave-dependent 35 approach yields a 25% larger surge over" Authors: The suggested revision has been made.

p15, line 14, "German Bight reaching a contribution of about 40% is some coastal area" -> "German Bight and reaching a contribution of about 40% in some coastal areas" Authors: The suggested revision has been made.

p15, line 15, "The contribution of the fully 3-D model in comparison with a 2D barotropic one yield to" -> "The contribution of a fully coupled 3-D model in comparison with an uncoupled 2-D barotropic one yielded up to"

45 Authors: The suggested revision has been made.

p15, line 23, "demand of disaster relief" -> "demands of disaster relief"

**Authors: The suggested revision has been made.**

**Reviewer #2**

**5**

10

In the new manuscript, more emphasis is given on the storm surge predictions from coupled -and uncoupled models, while the general experiment setup and case study are the same as in the previous paper. The advantage of having a coupled model system is discussed; the authors show that the coupling of their ocean circulation model with a wave prediction model improves the predictions of extreme storm

surges to a large degree. We are thankful....

We are thankful....

The relevance of wave-current interactions for storm surges still lacks sufficient documentation that is backed up with observational data, and this new paper presents strong arguments for using coupled models for the forecasting of dangerous storm surges. The data is presented clearly and informative in the figures, but the text needs some revision with regard to clarity and English grammar, therefore I would recommend the paper for acceptance with minor revisions.

**20 Authors: We completely agree and carefully revised our English grammar.**

Points to be corrected:

- Some references that are used in the text are missing in the reference list. Authors: We crossed-checked all references.

**25**

- page 2, line 8: wind-induced surface stress does generally play an important role, not only in shallow areas.

**Authors: We agree and rephrased this sentence.**

**30**

- page 2, line 11: The reference to Qiao et al (2004) is not an original reference to this problems, there are many earlier studies that treat wave-induced mixing in both experiments and models. It would be good to also cite some of the earlier works here.

**35 Authors: We cited earlier works and added new references.**

- page 6, line 12: If  is the sum of Eulerian current and the Stokes drift, equation (3) will solve for the Lagrangian current following water masses. This is somehow different to the way GETM solves for fixed grid points. If solving for  that includes Stokes drift, the radiation stress is not the only wave information that is used in eq. (3). Note that traditional formulations of radiation stress use a Eulerian framework. I think that that , as it is used here, should only include the Eulerian current.

Authors: We are sorry for the confusion. We completely agree with this statement and have made the appropriate corrections in the revised text.

45

40

The coupling from GETM to WAM should also be described along with section 2.3.

**Authors: We added this information to Section 2.3.**

- Some text passages, particularly section 6 are somehow hard to read and should be revised for clarity and grammar.

5

15

25

35

Authors: The text has been revised. The language and grammar have been corrected. We hope that the revised manuscript reads better.

**10 **Reviewer #3**

The manuscript presents a case study analysis of coastal inundation during an extreme extratropical storm event (Xaver) that made its land fall in northwest Europe December 2013), with a considerable impact in the North Sea. The central focus (and goal) of the paper is to show the goodness of having a wave model coupled to a surge model (or a regional ocean model).

I consider this study useful and interesting, nevertheless I have some comments regarding the way the study and the results are presented. I make some suggestions regarding language, but the authors should read and correct the whole text, since the use of the English language is sometimes far from appropriate.

20 Authors: We are very thankful for the suggestions regarding the language. We completely agree and carefully revised our English language and grammar.

Abstract The abstract has several flaws that I suggest the authors should address. Please have in mind that the abstract should "survive" (or stand) by itself. Hence it should have concise but complete information so that an educated reader knows (or understands) what to expect in the text body. Please provide information about the models you are using in the abstract.

Authors: We agree and added more information about the model and major results to the Abstract.

30 P1-L13 (same in L21 and L23): Extremes? What extremes? Extreme storm event? Extreme sea level rise?

Authors: This has been changed to "Extreme storm events".

P1-L21: replace "enhances significantly" with "is significantly enhanced" Authors: The suggested revision has been made.

P1-L23: replace "area" with "areas" Authors: The suggested revision has been made.

40 P2-L7: erase "the" before "ocean". Authors: This sentence has been revised for clarity.

P2-L7: Regarding sentence starting with "The wind-induced..." why is this here? It seams disconnected from the rest of the text (although, of course, being a valuable statement).

45 Authors: This statement has been revised for clarity.

P2-L8: sea surface or ocean surface (mixed) layer? I tend to look at the sea (ocean) surface as a skin layer. Please be clearer.

Authors: This statement has been revised for clarity.

P2-L13: add "a" before "circulation". 5 Authors: The suggested revision has been made.

P2-L16: I am afraid wave models are not earth system components. Regarding "... and further integrating of biogeochemical or morphologic parts" I don1t get what you mean; could you please re-phrase it? Authors: The suggested revision has been made and the statements re-phrased.

P2-L20: instead of "wind boundary layer" (which doesn't exist or it is not a valuable geophysical statement" please use "lower marine atmospheric boundary layer". All references here are from high wind speed regimes, when the highest (deeper) impact actually occurs during light winds and swell regimes.

Consider adding some references regarding light winds regime. 15 Authors: We added references regarding weak wind regimes to the introduction.

P2-L22 (and in several other parts of the text): add curly brackets on the years in the references. Authors: The suggested revision has been made.

20

10

P2-L26: what do you mean with "radiation stress approach"? Authors: We agree and this has been re-phrased in the revised manuscript.

P2-L27: what is a "practical analysis". I am afraid this might not be a very scientific statement. Authors: 25 This has been re-phrased in the revised manuscript.

P2-L28: what is "circulation for the ocean state"? Authors: The suggested revisions have been made.

30 P2-L30: the sentence starting with "The role of. . ." is lost here. No relation with before of after text. Authors: This sentence has been revised. This part is now the start of a new paragraph.

P3-L1: add "a" before "Lagrangian". "Drift" what drift? Stokes? Wave induced? Authors: The suggested revision has been made. It was also re-phrased making the description clearer.

35

P3-L3: replace semicolon with full stop and start new sentence afterwards. No need for this here (here and in other parts of the text).

Authors: The suggested revision has been made.

- 40 P3-L7: it is a fact that storm surges are meteorologically driven, not a "well accepted" situation. It would be the same as saying that "it is well accepted that ocean surface gravity waves are wind driven", or that "the thermohaline circulation is driven by water density differences". Authors: We completely agree and have made the suggested revision throughout the manuscript.
- 45 P3-L11: correct tense of sentence starting with "IPCC...". Authors: Following the suggestion of reviewer #1, we removed this paragraph from the introduction.

P3-15: please provide some more explanation on how waves and tides are amplified by the rise of sea level.

Authors: More information and explanations are provided including additional references

5 P3-L16: "could" or "can"? Authors: We changed to "can".

P3-L17: add "and" after "seawalls"; add "ocean" before "circulation". Authors: The suggested revision has been made.

10

P3-L18: add a comma after "Bight"; replace "greatest" with "great". Authors: The suggested revision has been made.

P3-L19: how can the forecast reduce the damage?Authors: The suggested revision has been made.

P3-L20: add "farms" after "energy"; replace "navigation" with "routing". Authors: The suggested revision has been made.

20 P3-L23: sentence starting with "Further. . ." is confusing; please re-write. Authors: The suggested revision has been made.

P3-28: what are external waves? Authors: We corrected this typo mistake.

25

P3- L30: replace "substantial" with "a considerable"; replace "for" with "in" Authors: The suggested revision has been made.

P4-L1: erase "cause".

30 Authors: The suggested revision has been made.

P4-L8: erase "as well as satellite data"; add "and remote sensing" after "in- situ". Authors: The suggested revision has been made.

35 P4-L22: "outer model" or "outer domain"? Authors: We corrected this in the revised manuscript.

> P4-L28: add "further details." after "2016)". Authors: The suggested revision has been made.

40

P5-L15: "action density" or "wave energy density"? Authors: We modified to "wave energy density".

P5-L21: there is no "S" in the rhs of equation (2).

45 Authors: The source terms  $S = S(\sigma, \theta, \varphi, \lambda, t)$  on the right hand side of the equation (2) is the net source term expressed in terms of the action density. It is tsplitted as he sum of a number of source terms representing the effects of wave generation by wind ( $S_{wind}$ ) quadruplet nonlinear wave-wave interactions ( $S_{nl4}$ ), dissipation due to white capping ( $S_{wc}$ ), bottom friction ( $S_{bot}$ ) and wave breaking ( $S_{br}$ ).

P6-L9: "wave motion" is too broad; please provide additional explanation

Authors: We agree and provided more explanation including additional references at the end of Section 5 2.4.

P6-L27: add "wave model" after "by"; the WAM model doesn1t "give" data!; all this sentence is inaccurate from a wave model standpoint.

Authors: The suggested revision has been made.

10

P7-L2: add "of" before "GOTM". Authors: The suggested revision has been made.

P7-L10: replace "causing" with "that caused".Authors: The suggested revision has been made.

P7-L19: replace "has" with "had". Authors: The suggested revision has been made.

20 P8-L19: erase double punctuation. Authors: erased.

P9-L5: replace "As an example we present" with "As can be seen in". Authors: The suggested revision has been made.

25

P9-L12: sentence starting with "The standard. . ." is confusing; please consider re-writing. Authors: The suggested revision has been made.

P9-L15: "low"?, how much?; replace "analysis on" with "the analysis of".Authors: "low" was substituted with a quantitative measure, the phrase has been replaced.

P9-L28: you have defined Hs before, hence erase "significant wave height". Authors: The suggested revision has been made.

35 P10-L28: replace "demonstrate" with "show"; maybe this reduction should be quantifies here. Authors: The suggested revision has been made. The reduction is quantified and this is demonstrated in Table 2.

P11-L14: reached or reaching?

40 Authors: This sentence has been revised for clarity.

P11-L30: replace "their" with "its". Authors: The suggested revision has been made.

45 P12-L3: erase "with". Authors: The suggested revision has been made. P12-L17: add "to be" before "important". Authors: The suggested revision has been made.

P12-L30: correct the tense of the verb.

5 Authors: The suggested revision has been made.

P13-L4: "North-Frisian Wadden Sea" is this correct? Authors: Yes the term "North-Frisian Wadden Sea" can be used also for the North Frisian Islands which a group of islands in the Wadden Sea, a part of the North Sea.

10

P13-L5: "is due. . ." how do you know? Authors: The suggested revision has been made.

P13-L6: replace "of the" with "to the".Authors: The suggested revision has been made.

P13-L29: models only could be inappropriate. . ." wrong tense; please re-write. Authors: The suggested revision has been made.

**20 P14-L2: replace "is" with "are". Authors: The suggested revision has been made.**

P14-L3: add "the" before Nederland". Authors: The suggested revision has been made.

25

P14-L6: sentence starting with "Recently. . ." is confusing; consider re-writing. Authors: The suggested revision has been made.

P14- L22: replace "with" with "to".

30 Authors: The suggested revision has been made.

P14-L30: replace "the coastal area" with "coastal areas". Authors: The suggested revision has been made.

35 P14-L30: replace "know" with "understand". Authors: The suggested revision has been made.

P14-L31: "risks and losses"? What do you mean?; replace "increases" with "has increased". Authors: This sentence has been revised for clarity.

40

P15-L1: sentence starting with "Although. . ." is confusing; consider re-writing. Authors: The suggested revision has been made.

P15-L4: what "leads"?

45 Authors: This sentence has been revised for clarity.

P15-L6: replace "has" with "have".

Authors: The suggested revision has been made.

P15-L7: replace "cause" with "causes". Authors: The suggested revision has been made.

5

P15-L9: erase "the". Authors: The suggested revision has been made.

P15- L18: sentence starting with "Nowadays. . ." is confusing; consider re-writing.
Authors: This sentence has been revised for clarity.

P15-L24: replace "satellite" with "remote sensing"; which products?, please be more specific. Authors: This sentence has been revised for clarity.

15 P15-L27: add "have after "We"; add "the" after "that". Authors: This sentence has been revised for clarity.

**Coastal flooding: impact of waves on storm surge during extremes. A case study for the German Bight**

5

**Joanna Staneva1, Kathrin Wahle1, Wolfgang Koch1, Arno Behrens1, Luciana Fenoglio-Marc2 and Emil V. Stanev1**

1. Institute for Coastal Research, HZG, Max-Planck-Strasse 1, D-21502 Geesthacht, Germany

2. Institute of Geodesy and Geoinformation, University of Bonn, Nussallee 17, D- 53115 Bonn, Germany

Correspondence to: J. Staneva (joanna.Staneva@hzg.de)

**10**

**Abstract**

This study addresses the impact of wind, waves, tidal forcing and baroclinicity on the sea level of the German Bight during extremes- storm events. The role of waveswave-induced processes, tides and baroclinicity is quantified, and the results are compared with observational data that include various in in situ measurements as well as and satellite data. A coupled, high-resolution, model modelling system is used to simulate the wind waves, the water level and the threedimensional hydrodynamics. The models used are the wave model WAM and the circulation model GETM. The two-way coupling is performed via the OASIS3-MCT coupler. The effects of the wind waves on sea level variability are studied, accounting for wave-dependent stress, wavebreaking parameterization and wave-induced effects on vertical mixing. The analyses of the coupled model results reveal a closer match with observations than for the stand-alone circulation model, especially during the extreme storm Xaver in December 2013. The predicted surge of the coupled model enhances is significantly enhanced during extremes extreme storm events when considering wave-current interaction processes. The wave-dependent approach yields to a contribution of more than 30% in some coastal area areas during extremes.extreme storm events. The contribution of a fully three-dimensional model compared with a twodimensional barotropic model showed up to 20% differences in the water level of the coastal areas of the German Bight during Xaver. The improved skill resulting from the new

developments justifies further use of the coupled-wave and three-dimensional circulation models for improvement of in coastal flooding predictions.

**1. Introduction**

A challenging topic in coastal flooding research is the provision of accurate predictions prediction of the sea surface elevations and wave heights. This is highly relevant over the European shelf-that, which is characterized by vast near-coastal shallow areas and a large nearcoastal urban population. The increased demand of improving to improve wave and storm predictions requires further development and improved representation of the physical processes in the ocean models. The wind-induced surface stress over the ocean plays an important role at shallow areain enhancing sea surface elevation (e.g., Flather, 2001). The importance of windwave-\_induced turbulence for the seaocean surface has been layer was demonstrated by Davies et al. (2000);), and it was further demonstrated for the bottom layer by Jones and Davies; (1998);) and for the wave-induced mixing by Babanin, (2006) and Huang et al. (2011). Craig and Banner (1994) and Mellor (2003) suggested that surface waves can enhance mixing in the upper ocean. Qiao et al. (2004) developed a parameterization of wave-induced mixing from the Reynolds stress induced by the wave orbital motion, and coupled this mixing towith a circulation model. They found that the wave-induced mixing can greatly enhance the vertical mixing in the upper ocean.

20 Understanding waves the wave-current interaction processes is important for the coupling between the different earth system components (e.g. ocean, atmosphere, wave and waves in numerical models) and further integrating of biogeochemical or morphological parts. Longuett. Longuet-Higgins and Stewart (1964) showed that wave-dissipation-induced gradients of radiation stress that accounts account for a transfer of wave momentum to the water column, 25 changing the mean water level. The effects of waves on the wind-lower marine atmospheric boundary layer arehave been demonstrated by a number of studies-like: Janssen (2004;), Donelan et al., (2012;), Fan et al., 2009. (2009), and for the light wind regimes: Veiga and Queiroz (2015); Sun et al., (2015). The effects of wave-current interactions caused by radiation stresses have been addressed by Brown and Wolf, (2009) and Wolf and Prandle, (1999-). A different approach, i.e., the vortex force formulation, was used by Bennis and Ardhuin (2011<del>);) and</del> McWilliams et al., (2004; Benetazzo), Kumar et al. 2013(2012). The

10

5

15

comparisonscomparison of both methods by Moghimi *et al.* (2013) showed that for the longshore circulations the results are similar, however for longshore circulations, but radiation stress enhanced the offshore\_directed transport in the wave shoaling regions is simulated using radiation stress approach.

- . Many other studies based on theoretical and practical analyses dealt withaddressed the role of the interaction between wind waves and circulation forin the ocean statemodel simulations (Michaud *et al.*, 2012, Barbariol *et al.* 2013; Brown *et al*7,2 2011; Benetazzo et al. 2013; Katsafados *et al.*, 2016; Bolaños *et al.*, 2011, 2014). The role of wave induced processes on Lagrangian transport and particle distributions -s demonstrated by, Röhrs et al. (2012, 2014). For idealized conditions, Weber et al, (2008) performed Lagrangian analysis of the mean drift due to dissipating surface gravity waves and showed that mean Lagrarian wave setup of the free surface and the mean drift solutions in a rotating ocean are given for a steady balanced flow; later Weber et al (2015) demonstrated by comparison with Lagrangian results that Coriolis Stokes force acts to change the vertically integrated Eulerian kinetic energy of the mean flow. The importance of waves current interaction of turbulence and bottom stress is shown by Babanin et al. (2010).
  - It is well accepted that a storm surge is meteorologically driven typically by wind and the atmospheric pressure. With further development of coastal economic and concentration of population in the coastal areas, dikes were built to protect from flooding. Even since, keeping dikes in order was a challenge, and keeping the dangers of storm surges at bay became an important issue (Storch, 2014). IPCC (2014) summarized that on regional scales it was very likely that there would be an increase in the occurrence of future sea level extremes in coastal regions by 2100 though and demonstrated the low confidence in region-specific projections in storminess and storm surges, thus more and more seawalls and levees would be overtopped in 2100 (Wang et al., 2012). Moreover, wave and tides would be amplified by the rise in sea level, which could increase the rate and intensity of this process, causing the collapse and damaging the seawalls/levees. Improved wave and circulation forecasts for the North Sea and its coastal areas, especially within the German Bight are of greatest importance for marine and coastal environment and can reduce the damages caused by flooding or coastal erosions. This is of utmost importance for example for the off-shore wind energy, ship navigations, and coastal zone protections. Surge and tidal combinations is known as "still water level" since in contrast to the very short wave period of 1-20 s, the periods of the water level vary from several hours to days.

15

25

30

Further superimposed by waves can additionally contribute to water level increase, causing further coastal flooding. Waves combined with higher water levels may break dykes, causing flooding and damaging and destroying constructions, coastal erosions (Pullen et al, 2007). The wave impact includes also wave breaking and changes of the beaches, increasing the coastal erosion and modifying the sediment dynamic (Grashorn et al., 2015; Lettman et al., 2009).

The German Bight area Storm surges are meteorologically driven, typically by wind and atmospheric pressure. As shown by Holleman and Stacey (2014), an increasing water level decreases the frictional effects in the basin interior, which alters tidal amplification. Waves combined with higher water levels may break dykes, cause flooding, destroy construction and erode coasts (Pullen *et al.*, 2007). Waves can also modify the sediment dynamics (Grashorn *et al.*, 2015; Lettman *et al.*, 2009).

The German Bight is dominated by strong north-westerly winds and external high waves due to the Northeast Atlantic low-pressure systems (Rossiter, 1958; Fenoglio-Marc et al., 2015). Extratropical cyclones in the area are substantial present a considerable hazard, especially forin the shallow coastal Wadden Sea areas (Jensen and Mueller-Navarra, 2008). The-Coastal flooding can be caused by the combined roleeffects of wind waves-together with, high tides and storm surges in response to fluctuations in local and remote winds and atmospheric pressure-cause. The role of thosethese processes can be assessed by using high-resolution coupled model coastal systems.models. However, in the frame of forecasting and climate modelling studies, the different processes of wave and current interactions are still not sufficiently exploited. In this study, we address the wave-current interaction processes in order to assess their the impact of waves on the sea level of the German Bight during extremes. We quantify their individual and collective role and compare the model results with observational data that include various in -situ and remote sensing measurements as well as satellite data... The wave model (WAM) and the), circulation model (GETM), the processes of their interaction, the study period as well as the differentand model experiments are presented in Section 2. The observational data that have been used are described in Section 3, followed by model-data comparisons in Section 4. Finally, Section 5 addresses the effects of the different physical processes on the sea level variability; followed by concluding remarks in Section 6.

5

15

25

20

**2. Models**

**2.1 HydrodynamicalHydrodynamic Model**

The circulation model used in this study is the General Estuarine Transport Model (GETM, Burchard and Bolding, 2002). The nested-grid model setup for the German Bight model set up has a horizontal resolution of 1 km and 21  $\sigma$ -layers- (Stanev et al., 2011). GETM uses the k- $\varepsilon$ turbulence closure to solve for the turbulent kinetic energy k and its dissipation rate  $\varepsilon$ . The open boundary data for temperature, salinity, velocity and sea surface elevation at the open boundary are obtained from the coarser resolution (eaapproximately 5 km and 21  $\sigma$ -layers) North Sea-Baltic Sea outerGETM model. Those models are described in details by configuration (Staneva et al. (., 2009); see also Fig. 1 for the bathymetry map). The sea surface elevation at the open boundary of the German Bightouter (North Sea-Baltic Sea) model domain.was prescribed using 13 tidal constituents obtained from the satellite altimetry via OSU Tidal Inversion Software (Egbert and Erofeeva, 2002). Both models were forced by are sea interaction atmospheric fluxes that are estimated using computed from bulk aerodynamic formulas. The atmospheric These formulas used model-simulated sea surface temperature, 2-m air temperature, relative humidity and 10-m winds from atmospheric analysis data-needed to estimate those fluxes are taken from the. This information was derived from the COSMO-EU regional model operated by the German Weather Service (DWD; Deutscher Wetter Dienst) and have), with a horizontal resolution of 7 km. The river run offRiver runoff data were provided by the German Federal Maritime and Hydrographic Agency (BSH, see also Staneva et al., 2016); Bundesamt für Seeschifffahrt und Hydrographie).

**2.2 Wave Model**

Ocean surface waves are described withby the two-dimensional wave action density spectrum  $N(\sigma, \theta, \varphi, \lambda, t)$  as a function of the relative angular frequency  $\sigma$ , wave direction  $\theta$ , latitude  $\varphi$ , longitude  $\lambda$  and time t. The appropriate tool to solve that the balance equation is the wellestablished advanced third-generation spectral wave model WAM (WAMDI group, 1988, ECMWF, 20152014). The use of the wave action density spectrum N is required if currents have to be are taken into account. In that case, the action density is conserved, in contrast to the energy density, which is normally used in the absence of time-dependent water depths and currents. The

10

5

15

action density spectrum is defined as the energy density spectrum  $E(\sigma, \theta, \varphi, \lambda, t)$  divided by  $\sigma$  observed in a frame moving with the ocean current velocity (Whitham, 1974, Komen *et al*3:1, 1994):

$$N(\sigma,\theta) = \frac{E(\sigma,\theta)}{\sigma} \underbrace{N(\sigma,\theta) = \frac{E(\sigma,\theta)}{\sigma}}_{(1)}$$

The wave action balance equation in Cartesian coordinates is given as:

$$\frac{\partial N}{\partial t} + (\mathbf{c}_{g} + \mathbf{U}) \nabla_{xy} N + \frac{\partial c_{\sigma} N}{\partial \sigma} + \frac{\partial c_{\theta} N}{\partial \theta} = \frac{S_{wind} + S_{nl4} + S_{wc} + S_{bot} + S_{br}}{\sigma}$$

$$\frac{\partial N}{\partial t} + (\mathbf{c}_{g} + \mathbf{U}) \nabla_{xy} N + \frac{\partial c_{\sigma} N}{\partial \sigma} + \frac{\partial c_{\theta} N}{\partial \theta} = \frac{S_{wind} + S_{nl4} + S_{wc} + S_{bot} + S_{br}}{\sigma}$$
(2)

The first term on the left-hand side of equation (2) represents the local rate of change of actionwave-energy density; the second oneterm describes the propagation of wave energy in the two-dimensional geographical space, where  $c_g$  is the group velocity vector and U is the corresponding current vector. The third term of the balance equation denotes the shifting of the relative frequency due to possible variations in depthsdepth and currentscurrent (with propagation velocity  $c_{\sigma}$  in  $\sigma$  space). And The last term on the left-hand side of the equation finally represents depth-induced and current-induced refraction (with the propagation velocity  $c_{\theta}$ in  $\theta$  space). The term  $S = S(\sigma, \theta, \varphi, \lambda, t)$  on the right-hand side of (2) is the net source term expressed in terms of the action density. It is the sum of a number of source terms representing the effects of wave generation by wind  $(S_{wind})$  quadruplet nonlinear wave-wave interactions  $(S_{nl4})$ , dissipation due to white capping  $(S_{wc})$ , bottom friction  $(S_{bot})$  and wave breaking  $(S_{br})$ . The current version of the thirdgeneration wave model WAM Cycle 4.5.4 is an update of the former Cycle 4-that, which is described in detail in Komen et al. (1994) and GuentherGünther et al. (1992). The basic physics and numerics are keptmaintained in that he new release. The source function integration scheme is provided by Hersbach and Janssen (1999), and the up dated updated source terms of Bidlot et al. (2007) and Janssen (2008) are incorporated. The Depth-induced wave model has breaking (Battjes & Janssen, 1978) is included as an additional source function. The depth and/or current fields can be non-stationary. The wave models have the same resolution, utilizes and the model uses the same bathymetry and wind forcing as the GETM model. The boundary values of the North Sea model are taken from the operational regional wave model of the DWD, while the

5

10

boundary values for the German Bight are taken from the North Sea model. The wave models run in shallow water mode, including depth refraction and wave breaking, and calculate the twodimensional energy density spectrum at the active model grid points in the frequency/direction space. The solution of the WAM transport equation is provided for 24 directional bands at 15° each with the first direction being 7.5°, measured clockwise with respect to true north, and 30 frequencies logarithmically spaced from 0.042 Hz to 0.66 Hz at intervals of  $\Delta f/f = 0.1$ .

**2.3 Coupled\_wave -circulation model implementation**

10 The implementation of the depth\_dependent equations of the mean currents  $\mathbf{u}(\mathbf{x}, \mathbf{z}, \mathbf{t})$  in the presence of waves follows Mellor (2011). Starting with The momentum equation for an incompressible fluid is  $d\mathbf{u}/dt = \mathbf{F} - \nabla \delta \mathbf{p}$ , where  $\mathbf{F}$  states for is the sum of external forces

20

5

(Coriolis, gravity, friction) and  $\nabla \delta p$  is the pressure gradient, which includes the influence of the wave motion on the mean current. Within the radiation stress formulation of Mellor (2011), the prognostic velocity u is related to the Eulerian wave-averaged velocity. Using linear wave theory and accounting for the second-order terms of the wave height, the equation of motions then readsmotion is:

$$\frac{\partial \langle \mathbf{u} \rangle}{\partial t} = \langle \mathbf{F} \rangle - \langle \mathbf{u} \rangle \cdot \frac{\partial \langle \mathbf{u} \rangle}{\partial \mathbf{x}} - \frac{\partial}{\partial \mathbf{x}} \cdot \mathbf{S} \underbrace{\frac{\partial \langle \mathbf{u} \rangle}{\partial t}}_{(3)} = \langle \mathbf{F} \rangle - \langle \mathbf{u} \rangle \cdot \frac{\partial \langle \mathbf{u} \rangle}{\partial \mathbf{x}} - \frac{\partial}{\partial \mathbf{x}} \cdot \mathbf{S},$$

where the angle brackets denote averaging over the wave period,  $\frac{1}{1000} - \frac{1}{1000} - \frac{1}$

the Eulerian current and the Stokes drift  $2\omega kH^2/(\sinh^2 kD)$  for waves with angular frequency  $\omega$ , wavenumber k and significant wave height  $H_s$  in a water column of depth  $D_{-}S$  is the radiation stress tensor:

$$\mathbf{S} = E\left(\frac{c_f}{c_g}\left[\frac{\mathbf{k}\otimes\mathbf{k}}{k^2} + \delta\right]\frac{\sinh 2kh + 2kh}{\sinh 2kD + 2kD} - \delta\frac{\cosh 2kh - 1}{4\sinh^2 kD}\right),\\ \mathbf{S} = E\left(\frac{c_f}{c_g}\left[\frac{\mathbf{k}\otimes\mathbf{k}}{k^2} + \delta\right]\frac{\sinh 2kh + 2kh}{\sinh 2kD + 2kD} - \delta\frac{\cosh 2kh - 1}{4\sinh^2 kD}\right),\tag{4}$$

with where  $E = 1/16gH_s$  is the wave energy, k is the wave vector k, and  $h = D(1 + \xi)$  is the local depth of layer  $\xi$ . Thus, the divergence of the radiation stress is the only (to

second order) force related to waves in the momentum equations. The equation for kinetic energy, which is derived from the momentum equation by multiplication with the velocity vector, reads as is:

$$\frac{\partial E_{kin}}{\partial t} = \langle \mathbf{F} \rangle \cdot \langle \mathbf{u} \rangle - \langle \mathbf{u} \rangle \cdot \frac{\partial E_{kin}}{\partial \mathbf{x}} - \frac{\partial}{\partial \mathbf{x}} \cdot \mathbf{S} \cdot \langle \mathbf{u} \rangle,$$
(5)

- where the gradients in wave energy (*i.e*-., dissipation due to wave breaking) may lead to increased surface elevationselevation (wave setup).
- The necessary-wave state information required to account for the divergence of the radiation stress in the GETM momentum equations is provided by WAM. WAM gives also data onThe dissipation source functions (wave breaking and white capping, as well as bottom dissipation) to estimated by the wave model WAM are also used in the turbulence module of GOTM. There it isThese data are used forto specify the calculation of boundary conditions for the dissipation of the turbulent kinetic energy and the vorticity due to wave breaking and due to bottom friction (Pleskachevsky *et al.*, 2011). Additionally, bottom friction depending on) Following Moghimi *et al.* (2013), an enhanced bottom roughness length  $z^{b_0}$  is computed as a function of the base roughness  $z_0$  and wave properties) have been implemented ( (e.g., the bottom orbital velocity of the waves) according to Styles and Glenn, (2000). This allows accounting for the generated turbulence at the bottom due to the non-resolved oscillating wave motion. In the two-way coupling experiments, the GETM model provides the water level and ambient current to WAM.

The coupling between GETM and WAM is performed via the coupler OASIS3-MCT: Ocean, Atmosphere, Sea, Ice, and Soil model at the European Centre for Research and Advanced Training in Scientific Computation Software (Valcke *et al.*, 2013). The name OASIS3-MCT is a combination of OASIS3 (Ocean, Atmosphere, Sea, Ice, and Soil model coupler version 3) at the European Centre for Research and Advanced Training in Scientific Computation (CERFACS) and MCT (Model Coupling Toolkit), which was developed by Argonne National Laboratory in the USA. The details of the properties and use of OASIS3 can be found in Valcke (2013). The exchange time between models is five minutes. This small coupling time step is a major advantage for modelling fastmoving storms compared to off-line (without using a coupler) coupled models, as in Staneva *et al.*, (2016), where hourly wave fields are used in GETM.

5

10

15

25

30

**2.4 Study period (meteorological conditions)**

This study is focused on the period during the winter storm Xaver that occurred on the 5th and 6th of December, 2013, causing and caused flooding and serious damages indamage to the southern North Sea coastal areas. During 4th to 7th of December, the storm depression Xaver moved from the south of Iceland over the Faeroe IslandFaroe Islands to Norway and southern Sweden and further over the Baltic Sea to EstoniaLithuania, Latvia and LithuaniaEstonia. It hadreached its lowest sea level pressure on 5the 5th of December at 18 UTC over Norway (ca.approximately 970 hPa, FigureFig. 2 and 3). It is interesting to notice here that Over the German Bight-area, the stormarrival of Xaver coincided with high tides-and thus; therefore, an extreme weather warning was given to the coastal areas of north-western Germany due to morehigh tides and wind gusts of greater than 130 km/h recorded wind gusts (Deutschländer et al., 2013). The extremely high water level and waves triggered sand-displacements displacement on the barrier islands and erosion of dunes in the Wadden Sea regionsregion. The German Weather Service reported the storm to be worse or similar to what has been experienced during the North Sea flood of 1962, in which 340 people lost their lives in Hamburg, saying that improvements in sea defences since that time would withstand this the storm surge (Deutschländer et al., 2013, Lamb and Frydendahl, 1991).

**20**

**2.5 Numerical experiments**

For the control simulation (CTRL run), GETM is run as a fully three\_dimensional baroclinic model without a coupling with the wave model as described in Section 2.1.. The fully two way coupled GETM WAM model simulations account for the processes as described in Section 2.3. The coupling is performed via the coupler OASIS3 MCT: Ocean, Atmosphere, Sea, Ice, and Soil model at the European Centre for Research and Advanced Training in Scientific Computation Software, (Valcke et al., 2013).

The effects onof using different coupling methods are studied by comparing the two-way fully coupled GETM-WAM model simulationssimulation (FULL run) with the one-way coupled model, in which the circulation model obtains information from the wave model-WAM (one-way coupling); we name). We denote this experiment further as FORCED run. Additionally, we run

5

10

the circulation model GETM as a two-dimensional barotropic model (so called 2D2-D run). In the last experiments final experiment, we excluded exclude the rive run-offriver runoff forcing (NORIV run). The list of these experiments is given onin Table 1.

**5 **3. Observational data**

The tide gauge observations from the eSurge project (www.esurge.org) are used. An overview of the existing operational tide gauges in the North Sea and Baltic Sea regions are available at the webpages of the EuroGOOS regions NOOS (North West Shelf Operational Oceanographic System) and BOOS (Baltic Operational Oceanographic System), respectively: www.noos.cc and www.boos.org. The water level data used here wereare acquired through the NOOS ftp server.

The in -situ wave data are taken from the wave-buoy observational network operated in the North and Baltic Seas by the BSH: (http://www.bsh.de/de/Meeresdaten/Beobachtungen).

Additionally, for validation, we useduse satellite measurements of the significant wave height and sea level in the German Bight area derived from the Jason-2, CryoSaCryoSat-2 and SARAL/AltiKa altimetry satellite missions,. This last is here of special interest since the satellite passed over the North Sea at the time of stormduring Xaver. As explained in Fenoglio-Marc *et al.* (2015), the standard altimeter products wereare extracted from the Radar Altimeter Database System (RADS) (Scharroo, 2013). The sea water level corresponding to the instantaneous in – situ tide gauge measurement, andwhich was called Total Water Level Envelope (TWLE) in Fenoglio-Marc *et al.* (2015), has been setimated as the difference between the orbital altitude above the mean sea surface model DTU10 and the radar range corrected for the ionospheric and tropospheric path delay, solid Earth, sea state bias and load tide effects. Corrections for the ocean tide and for, the atmospheric inverse barometer effect and wind haveare not been applied. Further on, used. The storm surge has been set setimated by correcting the TWLE for the ocean tide given by the global ocean tide model GOT4.8 (Ray *et al.*, 2011), see Fenoglio-Marc *et al.* (2015) for more details.

**4. Model validation during the extreme storm surges**

20

25

In this section, we will analyse the wave model performance during Xaver using the FULL experiment. The significant wave heights  $(H_s)$  from the model simulations are in good agreement with the measured values. As an example we present can been seen in the time-series graph for stations-Elbe (top) and Westerland (bottom) stations, the measured  $H_s$  was greater than 7.5 m during 2-8 of December, 2013 (Fig. 4). During the extreme, the measured Hs, was above 7.5 m. It is noteworthy that the wave model over predicts The peak of Hs during the storm peak Hs is reached earlier in the model simulations compared to the observations (Fig. 4-b,4b, d), ). This could be due to the DWD winds which are over-estimated in storm conditions (Stanevawind data (see also Wahle et al., 2016). In addition, the maximum of the statistical wave height simulated by the model for the two locations (Fig. 4a, c) occurs earlier than the onethat of the measurements, which is due to the shifted maximum of the DWD wind forecasts. The standard deviation between the model and the measurements is 0.16 m for the Elbe and 0.12 m for the Westerland station. The correlation coefficients between the WAM simulations and measurements were always high above are greater than 0.9 for all stations, and the normalized RMS error wasis relatively low- (between 0.09 and 0.16 m). For the analyses on of the wave model performance, including different statistical parameters computed during the extreme event for all available German Bight stations, we refer to Staneva et al. (2016).

The wave spectra at the locations of FINO-1 and Elbe BSH buoy stations are given in Fig. 5 for the study period. The wave spectra from the model simulations (Fig. 5a, c) are in a good agreement with the spectra from the observations (Fig. 5a, c). The time variability of the spectral energy is wellaccurately reproduced by the model, and the energy around the peak is similar in the observations and simulations; however, the model patterns are smoother than the observed onespatterns.

In addition to the in -situ measurements, the satellite altimetricaltimetry data provide a unique opportunity to evaluate both the temporal and spatial variability simulated in the model along its ground-track at the time of the overflight of the German Bight, lasting aroundapproximately 38 sec (see Figure 6 aFig. 6a, b). The modelled significant wave height (*Hs*) varies along the satellite ground-tracks between 1.2 and 1.9 m during calm conditions on  $03.3^{\text{th}}$  of December, 2013 at 18:00 UTC (Fig. 6a), while for the period of extreme stormduring Xaver, *Hs* varies between 6.3 m and 9.4 m ( $06.6^{\text{th}}$  of December 2013 at 04:00 UTC, Fig. 6b). The spatial distribution of the wave model results*Hs* (Fig. 6c, d) is in good agreement with the satellite data in both cases. The latitudinal distribution of *Hs* simulated by the wave model (green dots) is

15

25

20

smoother than the onethat of the satellite data, which. This can be explained by the different way of post-processing of the satellite data of the significant wave height and by the statistical nature of its estimate by the model. For calm conditions (Fig. 6 c) 6c), Hs is slightly underestimated (ca.approximately 15 cm) in the coastal area and overestimated (ca.approximately 20 cm) in the open German Bight. During Xaver, the model slightly overestimates the satellite data in the open areas (with ca. 20-30 cm). These results are in good consistency consistent with the results shown inof Fenoglio-Marc *et al.* (2015)), who compared the SARAL data with the DWD wave simulations.
* * *
**4.2 Sea level and wave-induced forcing**

In this section, we will-demonstrate the performance of the hydrodynamic model to simulate the mean sea level and givepresent statistics obtained for the whole integrationstudy period. Detailed statistical analyses onof the model comparisons with measurements for the area of German Bight are quantified by Staneva *et al.* (2016), where the coupled model performance of the wave and hydrodynamical model results for the area of German Bight is shown to be in a good agreement with observations, not only during the calm conditions, but most importantly, during storm events. Therefore here, we will only provide new examples onof model-data validations, including also-satellite data that have not been used in the previous studies.

The geographicalgeographic representation of the bias between the model simulations and all available tide gaugesgauge data shows that the bias for most of the tide gauge stations is within the range of +/-0.1 m7 (Fig. 7). Exceptions are found in some coastal tide gauge data stations in the very shallow areas. This can be attributed to the relatively coarse spatial resolution (1 km) and consequently smoother model bathymetry in the shallow coastal waters. Storm surges are estimated by subtracting from the simulations and tide -gauge observations , storm surges were estimated by subtracting the ocean tide estimated using the T\_TIDE routine (Pawlowicz *et al.*, 2002). FromThe comparisons between individual simulations are only marginally affected by tidal simulation errors because the simulations share the same systematic tidal errors. Estimating the surge component, the direct influence of tidal simulation errors in over-tides is minimized because the surge signals from observations and model comparisons with the runs are derived by subtracting an individual estimate of the tidal signal for each dataset.

From the comparison between the surge model and satellite data shown on Figure (Fig. 8,), it can be concluded that the model results are in a very good agreement with the observations-both during. This holds for calm conditions (at  $03.12.3^{\text{th}}$  of December 2013), wherewhen the surge has relative small valueswas weak (less than 10 cm offshore and up to 25 cm near the coastal area, Fig. 8c) and most importantly,), as well as during the storm Xaver at  $06.12.\text{ on } 6^{\text{th}}$  of December 2013, wherewhen the surge reached almost 3 m. The statistics from the comparisons between the observations and the different experiments are presented on Tables 2. The results demonstrate that that Table 2. The coupling between circulation and waves improves significantly improves the surge predictions; oncewhen the effects of interactionthe interactions with waves are considered, both the bias and the RMSE are substantially reduced, (see Table 2).

The timetemporal evolution of the water level for the Helgoland tide gauge data (see Fig. 1 for its location) is shown in Fig. 9. The consistency between the model simulations from boththe CTRL and FULL runs is very good during normal meteorological conditions-; however, during the storm-event however, the water level simulated by the stand-alone circulation model is lower by aboutapproximately 30 cm compared tolower than the data from the Helgoland tidal gauge station. OnceWhen the wave-induced processes are taken into consideration\_considered, the simulated sea level (FULL run) is closer to approaches the observations. Including wave-current interaction processes improved interactions improves the root mean square errorserror and the correlation coefficient between the tide gauges data and the simulated sea level over the German Bight area (Table 2).

The surge height reaches values of aboutapproximately 2.5 m during Xaver-and has, with its maximum at low water. It is noteworthy that during theDuring Xaver-event, two surge maxima of the surge ( $S_{max1}$  and  $S_{max2}$  in green line Fig. 9) have been are observed. Fenoglio *et al.* (2015) described the first surge maximum as a wind-induced onemaximum. They noticed alsofound that, instead, at the Aberdeen and Lowestoft, stations, the surge derived from the tide -gauge records had only one maximum, It reached reaching the eastern North Sea coastal areas (anticlockwise propagation) about approximately ten hours later than in-Lowestoft (easternmost UK coast) and finally,), causing the second storm surge maximum that has been detected by the measurements in the German Bight-area. As shown inby Staneva *et al.* (2016)), the wave-induced mechanisms contribute to a persistent increase of the surge after the occurrence of the first maximum (with slight overestimation after the second peak). At the two maxima, the observed water level at the Helgoland tide gauge is in better agreement with the coupled model

15

(FULL run-the: black line) than with the CTRL simulated water level. The two maxima are underestimated by the stand-alone circulation model (CTRL-: red line), especially at high water, wherewhen the surge difference between the model results and the measurements is aboutapproximately 30 cm for the first peak and more than 40 cm duringfor the second peak (Fig. 9).

**5. Process studies**

**5.1 Sensitivity of surge predictions to coupling with waves**

10 In this section, we will analyse the role of wave-current interaction interactions in the storm surge model and will also demonstrate their the sensitivity to one-way versus two-way coupling. Fig. 10 shows the time series of the water heightlevel (black line) and the storm surge (red line) for six stations (see Fig. 1 for their locations) together with the differences of the surge between the FULL and CTRL runs (FULL-CTRL; green line) and the differences between the FULL and FORCED runs (FULL-FORCED; blue line). The surge during the extreme exceeds 2 m in the open-ocean stations and increases with up to 2.8 m in the proximity of near the coastal stations. The two separate storm surge maxima during the Xaver storm (as (described in Section 4) are elearly-seen inat the near-coastal stationsstation ST1-4, whereas for at ST6 (in the Elbe Estuary), the surge is keptremains at high values, even in the period between the two maxima. The 20 coupling with waves approach leads to a persistent increase of the surge, especially after the occurrence of the first maximum  $(S_{maxl})$ . The difference in the simulated surge between the FULL and CTRL runs (green line) reaches a maximum during the first peak of the surge and is substantial during the follow-upfollowing two days. For the Hörnum station (ST3), the raise ofincrease in the surge due to coupling with waves exceeds 35% compared to the CTRL data (Fig. 10c), At the north-easternmost station (ST4), the surge difference between the FULL and CTRL runs is moregreater than 70 cm, which gives results in a contribution of the wave-current interaction processes of moregreater than 40%. For the open-deeper open-water station (ST5, Fig. 10f), the maximum contribution is about approximately 30 cm-that makes, a 25% increase ofin the surge. The differences between the FORCED and FULL runs are relatively small (less than 4% of the total for all stations-, see the blue line). However, for the shallower Elbe Station (ST6, Fig. 10e), the effects of two-way coupling in comparisons compared to the FORCED run (one-way coupling) seemare important. Staneva et al. (2016) provided a summary of improved

5

model performance with respect to the prediction of the sea level, which is the main variable considered below in the analysis of extreme surges in the German Bight. The quantification of the performance shows that in a large number of coastal locations, both the RMS difference and the bias between the model estimates and observations are significantly reduced because of the improved representation of physics. Only in very few very near-costal tide gauge stations does the coupling not lead to improvements, which might be due to the insufficient resolution of the near-coastal processes in very shallow water regions.

- To giveprovide an illustration of the coastal impact caused by the storm Xaver, we analyse the horizontal patterns of the maximum storm surge (Fig. 11) over the four tidal periods T1-T4 (as specified in Fig 9). During the second peak (T3), the surge exceeds 2.8 m over the whole German Bight coast (Fig. 11c); the storm surge for thenear Elbe area is highergreater than 3m3 m. During the period of the first surge peak (T2, Fig. 11b), the maximum occurs in the Sylt-Römo Bight area (above 2.8 m) and along the Elbe and Weser estuaries (about approximately 2.5 m), however). Over the whole German Bight-area, the simulated surge is above exceeds 1.5 m. In the period of relativerelatively calm conditions before the storm (T1), the surge is relatively low (Fig. 11a, less than 30 cm). A decrease ofin the surge in directiontowards the north-western German Bight is simulated during T4 (Fig. 11d). The intensification of the storm surge from the open sea towards the coastal area is consistent with the specific atmospheric conditions during Xaver (Fig. 3).
- To better understand the impact of the processes of wave-circulation interactioncurrent 20 interactions on the surge simulations, we will also analyse the horizontal patterns of the maximum differences ofin the storm surge between the coupled model (FULL run) and the stand-alone GETM (CTRL run). The maximum differences for each grid point are estimated over the four tidal periods (Fig. 12, T1-T4). The patterns show that the differences between the FULL and CTRL runs during the first surge maximum are more noticeable atin the lowlyingvery shallow North -Frisian Wadden Sea. The maximum surge simulated by the fully coupled model exceeds the onethat of the CTRL run with caby approximately 60 cm along the Sylt-Römo Bight during the T2-period. The enhancement of the surge in the coastal area (see Fig. 11b) is may be due to the non-linear interactions processes nonlinear interaction between circulation and waves (the contribution of the wave-current interaction of the increase of the surge is moregreater than 25%) along the German Bight coastal region (Fig. 12a). For the T3 period, the maximum surge difference (of about approximately 55 cm) is concentrated along the

5

10

Elbe River-area; however-over the whole German Bight coast, the increase of in the surge due to wave-induced processes exceeds 40 cm-along the entire German Bight coast. During the second Xaver peak, the radiation stress contributes to a rise of the sea level along the whole German Bight coast and in direction to the, which is directed towards the Elbe-Weser river area. ForDuring the first peak (T1), the differences between FULL and CTRL run are more pronounced towardsnear the North Frisian Wadden Sea. The computed maximum surge differences are higher during the period T2 than the ones obtained during the period T3. For T4 (Fig. 12d)), the maximum difference of <del>caapproximately</del> 15 cm occurs for the east Frisian coast toward thetowards Elbe River area, whereas alongin the north-eastern partarea, the wave-induced processes do not contribute much to the mean sea level and the surge simulations imulations of the FULL runs are similar to the CTRL ones. run. The horizontal distribution of the patterns of Fig. 12 demonstrates the good consistency with the meteorological situation (Fig. 3). The effects of wave-induced forcing during the storm are also observednoticeable in the open North Sea (maximum surge differences are aboutapproximately 30 cm Fig. 12b, c) and are due to the dominant role of the radiation stress-even in the deeper areas, the differences between the FULL and CTRL surge estimates are moregreater than 20%. Even though Although the wave heights wereare much higher in the open sea, the water there is much deeper-and; thus, the differences in sea level between the FULL and CTRL runs are relatively small.

**20 **5.2 3-D versus 2-D barotropic models**

Depth-average\_averaged two-dimensional flow models are widely applied in storm surge simulation, and have been assumed to meet the requirementrequirements of the operational forecasts and of most. They are also widely used in many scientific studies. However, to study the flow characteristics of the storm surges, the use of only barotropic models <del>only</del> could be inappropriate, in particularis insufficient, especially in the large discharge estuaries. The flows in the surface and bottom layers are usually quite different, so that the depth-average-averaged two-dimensional modelmodels cannot sufficiently depict the flow structure. Even moreFurthermore, storm surge models do not account for the baroclinic processes, like thesuch as density-driven changes of thein water masses, which isare important in the estuarine environments.

30

The changes of in the sea level due to temperature for the Nederland coastal areas have been studied by Tsimplis *et al.* (2006). Dangendorf *et al.* (2013) showed that laterally forced steric

5

10

15

variation and baroclinic processes becameare important at decadal scales, while atmospheric forcing causes the annual variability ofin the sea level. Recently, Chen *et al* (2016, (2014) studied the role of the of-remote baroclinic and local steric effects toin the interannual sea level variability and found that thea three-dimensional model that consider considers the temperature and salinity can bettermore accurately simulate the changes in the water level related to the North Atlantic Oscillation (NAO) related changes of the water level. ). In these models, more realistic open boundary conditions (than in the barotropic models) are used that allow accountingto account for the dynamics of heat and salt. In this section,We quantify the rolebenefit of using a fully three-dimentionaldimensional model that also consider\_considers temperature and salinity<del>,</del> when simulating to simulate the sea level during extremes-will be analysed.

The patterns providing the surge differences between the FULL and 2D2-D runs confirm firstly that those differences are much larger during the storm Xaver (T2, Fig. 13b) than for theduring calm conditions (T1, Fig. 13a). For T2, the maximum difference growsincreases eastward from 2-5 cm at the western boundary of the German Bight to more than 80 cm along the North -Frisian Wadden Sea coast and near the Elbe and Weser estuaries. Those The surge differences decrease to 30 cm during the second peak of Xaver. After the storm, the three-dimensional effects contribute to an increase in the sea level in the direction of the Elbe Estuary (Fig. 13d). Those These effects can reach more than exceed 25% of the sea level increase in comparisons, compared to the 2D2-D model simulations (Fig7, 14). For the Elbe area, the 2D2-D model underestimates the mean sea level with about by approximately 1 m. This could cause significant underestimation of the sea level predictions performed by of the barotropic models-only. For T4, the impact of baroclinicity is localized along the south-eastern coastline (Fig. 13d). The differences between FULL and NORIV runs-(Fig. 14-, blue line) are negligible at station-ST3, while for theat ST6, they are about approximately 15 cm during storm Xaver in the vicinity of the Elbe Estuary during the storm Xaver. When analysing the impact of the baroclinicity on sea level model results, we use the barotropic hydrodynamic model that is not coupled to the wave model since our aim is to demonstrate only those effects. Introducing wave-circulation coupled processes (as demonstrated in the previous sections) to the barotropic model can reduce the differences between this model and the FULL run.

5

15

25

30

**6. Discussion and conclusions**

With the uncertainties of storm surge predictions under climate change, the quantification of associated hazardhazards is of great interest to the coastal area. areas. The demand to know what <del>changes inunderstand the risk of damage has increased for</del> the <del>risks and losses can be expected</del> when developing differentdevelopment of future projections increases. climate scenarios.

Although storm surge forecasting technology has been gradually mature, The accurate real-time assessment of the storm surgesurges and inundation areaareas is still defective unable to fully satisfy thethese demands because the real timeatmospheric storm forecasting, as the important driving force of surgesurges, is not perfect in practical use. It always. This leads to a high degree of uncertainty in storm surge forecasting until the final moment. The peak surge is depended depends on the accurate prediction of the landfall position and time. Since the astronomic tide may has a huge tide range in some coastal areas during the storm surge, the uncertainty of forecasting cause a dilemma in hazard relief. The future development of water level predictions will focus on both-enlarging the observation data network and further model developments. To reduce uncertainty, increasing the knowledge on theof various processes like, such as tide-wave-surge interactions, is needed. Improved weather forecast forecasts and also further coupling between the atmosphere, ocean and wave components will reduce the uncertainties.uncertainty. Increasing the horizontal resolution forin the near-coastal areas is made possible withby the availability of more computational resources. In this study, we show that the wave-dependent approach yields toa 25% larger surge atover the whole coastal area of the German Bight, reaching a contribution of aboutapproximately 40% isin some coastal areaareas during extremes. The contribution of The fully 3-D model in comparison with a 2D and the barotropic one yield to-model produce approximately 20% differences difference in the water level of the coastal areas of the German Bight during Xaver. The possible advantages of including the wave-current interaction in two dimensional barotropic models to improve sea level predictions will be the subject of further studies.

[revised manuscript text omitted]

15 We demonstrated that the interaction between waves and three-dimensional hydrodynamicalhydrodynamic models reduces the forecast errors, especially during extreme events. This will enable further use of the high\_resolution coupled model system for both improving the models to improve coastal flooding predictions as well as prediction and climate studies.

20

**Acknowledgements:**

This work has been supported throughby the Coastal Observing System for Northern and Arctic Seas (COSYNA) and as part of the Copernicus Marine Environment Monitoring Service (CMEMS) Wave2Nemo project. CMEMS is implemented by Mercator Ocean in the framework of a delegation agreement with the European Union. Luciana Fenoglio is supported by the European Space Agency (ESA) within the Climate Change Initiative (CCI). The authors are thankful tograteful for I. Nöhren for assistance with the graphics- and to-BSH for providing the observational data. The authors thank Sebastian Grayek for assistance with the tidal model.

5

| Experiment | 3-D GETM | WAM                          | Barotropic | Rivers run-off |
|------------|----------|------------------------------|------------|-----------------------|
| CTRL       | yes      | -                            | -          | yes                   |
| FULL       | yes      | Two-way                      | -          | yes                   |
| FORCED     | yes      | One-way                      | -          | yes                   |
| 2D         | -        | -                            | yes        | no                    |
| NORIV      | yes      | <del>yesTwo-way</del> | -          | no                    |

 Table 1. Model experiments.

5

|             | CTRL  | FULL  | FORCED | 2D    |
|-------------|-------|-------|--------|-------|
| RMSE        | 0.26  | 0.16  | 0.15   | 0.39  |
| Bias        | -0.17 | -0.09 | -0.10  | -0.28 |
| Correlation | 0.84  | 0.92  | 0.93   | 0.76  |

**Table 2.** Surge (m): Root-Mean Square Errors (RMSE), bias (model-observations) and correlation between storm surge component from four model runs (CTRL, FULL, FORCED and 2D) and from tide gauge records of the British Oceanographic Data Centre (BODC) over the German Bight area.

Figures